# W2GAN: Recovering an Optimal Transport Map with a GAN

## Abstract

Understanding and improving Generative Adversarial Networks (GAN) using notions from Optimal Transport (OT) theory has been a successful area of study, originally established by the introduction of the Wasserstein GAN (WGAN). An increasing number of GANs incorporate OT for improving their discriminators, but that is so far the sole way for the two domains to cross-fertilize. In this work we address the converse question: is it possible to recover an optimal map in a GAN fashion? To achieve this, we build a new model relying on the second Wasserstein distance. This choice enables the use of many results from OT community. In particular, we may completely describe the dynamics of the generator during training. In addition, experiments show that practical uses of our model abide by the rule of evolution we describe. As an application, our generator may be considered as a new way of computing an optimal transport map. It is competitive in low-dimension with standard and deterministic ways to approach the same problem. In high dimension, the fact it is a GAN-style method makes it more powerful than other methods.

## 1 Introduction

Generative Adversarial Networks (GANs) (Goodfellow et al., 2014) are powerful probabilistic generative models which have attained state-of-the-art results, especially in high-dimensional image data. Recently, Optimal Transport (OT) theory has contributed to the success of GANs, mainly in defining more robust training objectives (Arjovsky et al., 2017; Gulrajani et al., 2017; Miyato et al., 2018; Sanjabi et al., 2018). In this work, we explore a contribution in the other direction: we show that an adversarially trained generative model can recover an *optimal map*, i.e. a map between two distributions with a minimal cost of "transport".

Estimation of optimal maps constitutes a central problem in the OT community, especially for high-dimensional continuous distributions (Peyré & Cuturi, 2018). The well-known success of GANs in generative modelling of high-dimensional data suggests that such a generator might provide a good approximation of the optimal transport map in high-dimensional applications. There has also been considerable recent interest in using GANs to learn unsupervised mappings across domains (Zhu et al., 2017; Almahairi et al., 2018; Galanti et al., 2018). However, to date, these methods generally rely on architectural features and heuristics to ensure meaningful mappings. OT theory provides a means of formalizing learned mapping between distributions, hence placing this line of inquiry on stronger theoretical foundations (Courty et al., 2017a).

A fundamental problem with existing GAN frameworks is that we cannot theoretically characterize the training dynamics of the generator, nor the final mapping that it learns. As the generator's training signal derives from the discriminator, any formal description of the generator's training dynamics must rely on a theoretical characterization of the discriminator throughout training. In the case of Wasserstein GANs (Arjovsky et al., 2017) and its extensions, there is no uniqueness of the optimal discriminator at each update step, and hence the generator may follow one of infinitely many paths of evolution throughout training (Arjovsky & Bottou, 2017). For traditional GANs relying on $f$-divergences, discriminators are updated only a few times at each iteration of the training procedure, and could be far from the optimal discriminator. In such cases, characterizing the generator's evolution in terms of the ideal discriminator is irrelevant. On the other hand, the uniqueness of optima and the

simplicity of geodesics in the case of the second Wasserstein metric (Villani, 2008) makes it a good candidate as a discriminator objective.

We introduce the W2GAN that minimizes the second Wasserstein distance. Within the W2GAN framework, we can characterize the evolution of the generated distribution. More specifically, we show that it follows the Wasserstein-2 geodesic between the generator's initial distribution and the target distribution. Crucially, the generator of W2GAN is not learning an arbitrary mapping onto the target distribution. We show that, at convergence, it is reproducing the optimal transport map between the generator's initial distribution and the target distribution.

Our contributions are the following:

- We introduce W2GAN, in which the generator recovers an optimal map in high-dimensional spaces.

- We show that the training dynamics of our generator is characterized as following a unique geodesic between the initial distribution and the target distribution. Our analysis is based on interpreting the signal given by the discriminator at each update of the generator as a local update toward an optimal transport map. While this can be applicable to other Wasserstein GAN models, with the W2GAN, this insight extends to a global description of the generator's trajectory.

- We verify our proposed model on synthetic low-dimensional data, and show that it perform competitively on high-dimensional image data.

## 2 RELATED WORK

The intersection of GANs (Goodfellow et al., 2014) and OT theory was mainly initiated by the introduction of WGAN (Arjovsky et al., 2017), which proposed using the first Wasserstein distance as an alternative to the $f$-divergence (Nowozin et al., 2016). The objective of the discriminator in WGAN and its extensions (Arjovsky et al., 2017; Gulrajani et al., 2017; Miyato et al., 2018) corresponds to a particular instance of the Kantorovitch problem, which is the main problem at stake in OT theory. The Kantorovitch problem consists of finding an optimal plan for "transporting" one distribution onto another according to a predefined point-wise cost function. Borrowing these notions from OT theory has proved to be useful for improving both the understanding and performance of GANs. Salimans et al. (2018) proposed replacing the discriminator with the Sinkhorn algorithm, which can compute any Wasserstein metric. Alternatively, Sanjabi et al. (2018) used the optimal transport map as a new way to train a GAN discriminator.

The optimal map has other important applications such as computing barycenters of distributions (Rabin et al., 2011; Bonneel et al., 2015). Wasserstein barycenters may be used as a meaningful way to interpolate between probability measures (Cuturi & Doucet, 2014). Nonetheless, computing the optimal map remains challenging in some situations. Recently, regularized versions of the Kantorovitch problem were proposed to provide improved algorithms for computing the optimal map between (semi-)discrete distributions (Cuturi, 2013; Genevay et al., 2016). Seguy et al. (2018) developed a parametric approach to obtain a two-step procedure for computing an optimal map in the case of continuous distributions. In all of these methods, dealing with high dimensional data is challenging. One naturally suspects that a GAN-based approach of this problem would be efficient, knowing the success of such models in high dimensions.

For learning an optimal map using GANs, Wasserstein metrics are of main interest, especially given the relationship between the discriminator and the optimal map (Lei et al., 2017). In this context, however, the first Wasserstein distance is not convenient as the gradient flow of this metric is hard to solve (Ambrosio et al., 2008), and the relationship between the discriminator and the optimal map is neither unique, nor easily computed. What misses is a unique and tangible characterization of the training dynamics of the generator.

Characterizing generator's dynamics is a complex problem. We need to understand the evolution of a probability measure through a signal given by a real function over distributions – the discriminator. OT theory deals extensively with the characterization of paths in the space of probability measures. Specifically, it allows one to relate probability measures and their evolution through gradient flows (Villani, 2008; Ambrosio et al., 2008). Bottou et al. (2018); Arjovsky & Bottou (2017)

explored how to relate the generator's evolution with paths in the space of probability measures described by the discriminator. In the particular context of Wasserstein-based GANs, it is possible to link the generator's evolution with the Wasserstein-1 geodesics, which unfortunately happen to be in a great number. On the other hand, in the case of the second Wasserstein distance, the geodesics are unique and related to the optimal map in a simple way (Ambrosio et al., 2008).

## 3 BACKGROUND

We state in this section some basic notions from OT theory (Villani, 2008; Santambrogio, 2010; Ambrosio & Gigli, 2013) which will be necessary for the rest of the paper. We provide a more comprehensive presentation in Appendix 8.4.

**Optimal Transport problem and Wasserstein distances**   Computing Wasserstein distance and optimal transport map are two intrinsically related problems. Given two probability measures $\mu$ and $\nu$ on the euclidean space $\mathbb{R}^d$, an *optimal map* or a *Monge map* $T$ is a function minimizing the Monge problem:

$$\inf_{T \in \mathcal{A}_T} \int_{\mathbb{R}^d} c(x, T(x)) d\mu(x) \tag{1}$$

where $\mathcal{A}_T$ is the set of all maps from $\mathbb{R}^d$ to $\mathbb{R}^d$ that respect the marginal $\nu$, or more formally written as $T_\# \mu = \nu$ [1]. This objective involves a fixed point-wise cost function $c : \mathbb{R}^d \times \mathbb{R}^d \to \mathbb{R}$. When the cost $c$ is a power $p$ of the euclidean distance, the value of (1) is called the $p$-th Wasserstein distance, and denoted $W_p(\mu, \nu)$ [2]. We will assume this form for the cost function in the rest of the paper. In this case, it can be shown that we recover the value of $W_p(\mu, \nu)$ with a relaxed formulation of (1), which is called the *dual Kantorovich problem*:

$$\sup_{\phi, \psi \in \mathcal{A}^*(\mu,\nu)} \int_{x \in \mathcal{X}} \phi(x) d\mu(x) + \int_{y \in \mathcal{Y}} \psi(y) d\nu(y) \tag{2}$$

$$\mathcal{A}^*(\mu, \nu) := \{(\phi, \psi) : \mathbb{R}^d \to \mathbb{R} / \; \forall x, y \in \mathcal{X} \times \mathcal{Y}, \phi(x) + \psi(y) \leqslant c(x, y)\}$$

We will also denote as $V^*(\phi, \psi)$ the value $\int_{x \in \mathcal{X}} \phi(x) d\mu(x) + \int_{y \in \mathcal{Y}} \psi(y) d\nu(y)$. A pair $(\phi, \psi)$ maximizing 2 are called *Kantorovitch potentials*. We will refer to the constraint in the definition of $\mathcal{A}^*(\mu, \nu)$ as the *c-inequality constraint* or simply the *inequality constraint*.

In WGAN and its extensions, the discriminator corresponds to these Kantorovitch potentials computing $W_1$. More generally, it is possible to consider $\phi, \psi$ as discriminators computing any Wasserstein distance $W_p$ (Salimans et al., 2018; Sanjabi et al., 2018). On the other hand, the generator's update rule is driven by the gradient of the discriminator, i.e. the Kantorovitch potentials. In the particular case of of $W_2$, we can derive a simple relationship between these potentials' gradients and the Monge map $T$ solving (1).

**Relating Potentials to the Monge map**   An important property, which holds for $W_2$, is that we can relate the dual Kantorovitch solution $\phi$ and the Monge map $T$ (Brenier, 1991). $T$ is unique and determined by:

$$T = Id - \nabla \phi \tag{3}$$

where $Id$ is the identity. A detailed and more generalized statement of this relationship is given by Proposition (2) in the Appendix. In fact, there is also a converse statement: given a function $T : \mathbb{R}^d \mapsto \mathbb{R}^d$ and a probability measure $\mu$, $T$ is a Monge map between $\mu$ and $T \# \mu$ if $T$ is the gradient of a strictly convex function. That is, If $T$ can be written as $x - \nabla \phi(x)$ for some real function $\phi$ one can rephrase this condition as $\frac{\|x\|_2^2}{2} - \phi(x)$ being strictly convex. Such a $\phi$ is said to be *c-concave*.

This relationship suggests that when the discriminator is optimizing the $W_2$ metric, the generator would be driven by the optimal map $T$.

---

[1] The map $T_\# \mu$ is called the *push-forward measure* of $\mu$ and is defined as $T_\# \mu(A) = \mu(T^{-1}(A))$ for any Borel set $A \in \mathbb{R}^d$.

[2] Rigorously, optimizing (1) leads to $W_P^p(\mu, \nu)$

## 4   W2GAN: THE MODEL

In this section, we introduce the W2GAN model in which the discriminator computes the second Wasserstein distance $W_2$. Let us denote our discriminator $(\phi, \psi)$, a target distribution $P_x$ and our generator $G(z)$ where $z \sim P_z$. Here $z$ is not necessarily meant as the usual low-dimensional latent variable in GAN. In all generality, $z$ could have the same dimension as $x$ (e.g. as in unsupervised domain translation applications). In such a case, the next section will show that if $G$ is initialized as the identity function, the training of our generator will compute a Monge map between $P_z$ and $P_x$.

For optimization purposes, it is preferable to deal with a regularized version of (2). We choose to use the $L_2$ regularization, but one could use other penalties (Cuturi, 2013; Seguy et al., 2018; Blondel et al., 2017) [3]. Hence our discriminator's objective for approaching $W_2^2(G \# P_z, P_x)$ is:

$$\sup_{\phi, \psi} \mathcal{L}_D(\phi, \psi, G) := \sup_{\phi, \psi} \mathbb{E}_{(z,x) \sim P_z \times P_x} \left[ \phi(G(z)) + \psi(x) - \lambda_{\text{ineq}}(\phi(G(z)) + \psi(x) - \frac{\|G(z) - x\|_2^2}{2})_+^2 \right] \tag{4}$$

where $(.)_+ := \max(0, .)$ and $\lambda_{\text{ineq}}$ is a scalar controlling the strength of the penalty. We will refer to the penalty term by $\mathcal{L}_{\text{ineq}}(\phi, \psi)$ in the rest of the paper. Our generator's objective will then be

$$\inf_G \mathbb{E}_{z \sim P_z}(\phi(G(z)) \tag{5}$$

The training procedure of W2GAN is detailed in Algorithm 1. In the next section we will prove that W2GAN's generator can recover the Monge map between the initial generated distribution and the target distribution. The experiment section will provide empirical evidence confirming this and comparing it with other methods.

## 5   THE GENERATOR RECOVERING A MONGE MAP

The proof that our generator recovers a Monge map consists of characterizing the dynamic of the W2GAN's generator during training. We observe that it follows a pre-determined trajectory: the unique $W_2$ geodesic between the initial distribution and the target distribution. This result is stronger than only recovering the Monge map at the end of training, which will be a corollary to the analysis of the $W_2$-geodesics detailed in the background section. Hence, this extra piece of information is another interesting theoretical fact: one can uniquely characterize the dynamic of evolution of our generator during training. The analysis of this evolution is divided in two part. First we examine the direction of the update in the space of probability measure locally, for one generator update. Second, we concatenate these local updates to obtain a global description of the trajectory. The first technical tool we need is the characterization of geodesics in the $W_2$ case.

**Geodesics in the space of probability measures**   Recall that $W_2$ is a metric over distributions and thus makes the space over probability measures a metric space (Villani, 2008). Studying properties of this space starts with the analysis of its geodesics. That is, for any given distributions $\mu$ and $\nu$, we look for paths described by $\mu_t$ such that $\mu_0 = \mu$, $\mu_1 = \nu$ and

$$\forall 0 \leqslant s \leqslant t \leqslant 1, W_2(\mu_s, \mu_t) = (t - s)W_2(\mu, \nu)$$

It happens that such geodesics are unique (Villani, 2008). Given the unique Monge map $T$ between $\mu$ and $\nu$, the only constant speed geodesic is given by $\mu_t := T_t \# \mu$ where $T_t = (1 - t)Id + tT$. By (3), we can write $T(x) = x - \nabla\phi(x)$ for any Kantorovitch potential $\phi$ solving (2). Hence $T_t(x) = x - t\nabla\phi(x)$. In particular, this guarantees that $t\phi(x)$ is a Kantorovitch potential solving (2) for $\mu$ and $\mu_t = T_t \# \mu$.

We now move to the analysis of our generator's training dynamics. We first discuss the general update rule for any GAN model.

---

[3]The same regularized approach is taken for the gradient penalty or lipschitzness of the discriminator in Gulrajani et al. (2017) and Petzka et al. (2018). This lipschitzness constraint is in fact a particular case of the general $c$-inequality constraint in the case of $W_1$ (Villani, 2008). Hence WGAN-GP and its extensions rely on the exact same procedure: softening the hard constraint by adding a penalty in the discriminator's objective

**Local evolution of the generator of any GAN**  Let us consider the general objective of a GAN

$$\min_\theta \max_D E_{z \sim P_z} \mathcal{L}_1(D, G_\theta(z)) + E_{x \sim P_x} \mathcal{L}_2(D, x) \tag{6}$$

We can also use the random variables notations $G_\theta(z) \sim G_\theta \# P_z =: \mu_\theta$ and $x \sim P_x$. Denoting $\theta_t, \theta_{t+1}$ the current parameter for the generator and the next one and $\alpha > 0$ the hyper-parameter update, the above leads to the common update rule used by GAN methods:

$$\theta_{t+1} = \theta_t - \alpha \nabla_{\theta_t} E_{z \sim P_z} \mathcal{L}_1(D, G_{\theta_t}(z)) = \theta_t - \alpha E_{z \sim P_z} \nabla_{\theta_t} \mathcal{L}_1(D, G_{\theta_t}(z)) \tag{7}$$

where the last equality holds when one can control the norms of the gradients in order to invoke the Dominated Convergence theorem. We reinterpret this update rule using the functional gradient analysis as in (Anonymous, 2019; Johnson & Zhang, 2018). One may see that we obtain exactly the same update rule by differentiating the following objective at $\theta = \theta_t$:

$$\min_\theta \mathbb{E}_{z \sim P_z} \|F_\alpha(G_{\theta_t}(z)) - G_\theta(z)\|_2^2 \tag{8}$$

where $F_\alpha(x) := x - \frac{\alpha}{2} \nabla_x \mathcal{L}_1(D, x)$. Hence it is a completely equivalent formulation of the minimization objective for the generator. Eqn. (8) states that given the current generated distribution $G_{\theta_t}(z)$, the next parameter $\theta$ will be chosen so that $G_\theta(z)$ will be close to $F_\alpha G_{\theta_t}(z)$. Then if we want to locally understand how the generated distribution moves, we have to analyze this push-forward by $F_\alpha$. Instead of looking at the space of parameters, we directly consider the space of probability measures. Given the current $\mu_{\theta_t}$, we consider the multiple possible push-forwards of $\mu_{\theta_t}$, that is all the $H \# \mu_{\theta_t}$ for different functions $H$ from $\mathbb{R}^d$ to itself. One may also see a push-forward as the random variable $H(G_{\theta_t}(z))$. As in (Anonymous, 2019), we associate a loss to any such function:

$$\mathcal{L}(H) := \mathbb{E}_{z \sim P_z}(H(G_{\theta_t}(z))) \tag{9}$$

In particular, $\mathcal{L}(Id)$ is just the current loss of the generator. We can take the derivative of $\mathcal{L}$ at identity to obtain the direction onto which the generated distribution should be updated, but now in the space of probability measures instead of the space of parameters. This is done by computing Gateaux differentials. That is, we first consider any direction $h$ and compute the following directional derivative assuming bounds and regularity which allows us to switch the order of the expectation and the derivative:

$$\lim_{\epsilon \to 0} \frac{\mathcal{L}(Id + \epsilon h) - \mathcal{L}(Id)}{\epsilon} = \mathbb{E}_{z \sim P_z}[\nabla \mathcal{L}_1(D, G_{\theta_t}(z)) h(G_{\theta_t}(z))] \tag{10}$$

Equation (10) is homogeneous in $h$ and thus the gradient of $\mathcal{L}$ at identity is the function $\nabla \mathcal{L}_1(D, x)$. This is tantamount to saying that choosing $h(x)$ to be $-\nabla_x \mathcal{L}_1(D, x)$ will be locally the best choice for decreasing the loss of the generator. We thus consider the update in the space of distributions that sends $Id(G_{\theta_t}(z)) = G_{\theta_t}(z)$ onto $H_\alpha(G_{\theta_t}(z))$ where $H_\alpha(x) := x - \frac{\alpha}{2} \nabla_x \mathcal{L}_1(D, x)$. This is analogous to gradient descent in the space of probability measures: we considered the loss of the current distribution $Id(G_{\theta_t}(z))$ and updated it in the most significant direction given by $H_\alpha$. Now, we can observe that $H_\alpha = F_\alpha$. Let us summarize what this means: given the current loss of the generated distribution $\mu_{\theta_t}$, the update rule in the ideal space of probability measures consists of taking the new distribution $F_\alpha \# \mu_{\theta_t}$. Then the update rule according to (8) in the space of parameters is only choosing $\theta_{t+1}$ so that the new generated distribution $\mu_{\theta_{t+1}}$ is as close to the ideal update $F_\alpha \# \mu_{\theta_t}$.

**Links with a certain Monge map**  It is tempting to interpret the previous $F_\alpha(x) = x - \frac{\alpha}{2} \nabla_x \mathcal{L}_1(D, x)$ as an optimal transport map (or Monge map) in the Wasserstein-2 sense between the current distribution $\mu_{\theta_t}$ and the distribution $F_\alpha \# \mu_{\theta_t}$. Indeed, the expression of $F_\alpha$ is quite similar to the Monge map $T$ given in proposition (3). That is, $T(x) = x - \nabla \phi(x)$ where $\phi$ is a Kantorovitch potential maximizing (2). However, there are still two apparent obstacles:

1. In the expression of $F_\alpha$ the scalar $\alpha > 0$ has no correspondence in the expression of $T$.
2. How can we ensure that $\mathcal{L}_1(D, x)$ is indeed a Kantorovitch potential, i.e. is an optimum of (2) for $\mu = \mu_{\theta_t}$ and $\nu = F_\alpha \# \mu_{\theta_t}$

The second question is answered in the background section: the requirement that $\mathcal{L}_1(D, .)$ needs to be $c$-concave. This does not hold for general GANs where the complexity of the discriminator function makes it impossible to decide whether $\frac{\|x\|_2^2}{2} - \mathcal{L}_1(D, x)$ is strictly convex. In our case,

$\mathcal{L}_1(D, x) = \phi(x)$ where $\phi$ is a Kantorovitch potential. A Kantorovitch potential automatically satisfies such a condition, so the problem is solved.

For the first concern, when $\alpha = 2$, $F_2$ is exactly the Monge map $T$ from $\mu_{\theta_t}$ to $P_x$. The characterization of $W_2$ geodesics in (5) shows that any $F_\alpha$ for $0 \leqslant \alpha \leqslant 2$ is actually also a Monge map between $\mu_{\theta_t}$ and $F_\alpha \# \mu_{\theta_t}$. This solves the first obstruction. We also know that $F_\alpha \# \mu_{\theta_t}$ lies in the unique Wasserstein 2 geodesic between $\mu_{\theta_t}$ and $F_2 \# \mu_{\theta_t} = P_x$. We can summarize this by saying that an update of the current generated distribution $\mu_{\theta_t}$ consists of taking a step of size $\alpha > 0$ along the optimal transport map toward the distribution $\mu_{\theta_t} - \nabla \mathcal{L}_1(D, G_{\theta_t}) \# \mu_{\theta_t} = \mu_{\theta_t} - \nabla\phi \# \mu_{\theta_t} = T \# \mu_{\theta_t} = P_x$. In the experiment section, we empirically confirm that the discriminator provides a signal along the optimal map.

**From local to global dynamic of the generator**    We now want to extend the local description of the generator's dynamics by showing that the subsequent generated distributions $\mu_{\theta_0}, \mu_{\theta_1}, \ldots \mu_{\theta_n}$ all live on the $W_2$ geodesic joining $\mu_{\theta_0}$ to $P_x$. Let us assume this is the case for a fixed $n > 0$. As in the previous paragraph, we will assume the parametric update to match the update in the space of probability measures [4]. From the previous paragraph, the update at time $n$ in the space of probability measures leads to $\mu_{\theta_{n+1}} = F_\alpha \# \mu_{\theta_n}$. From the notations introduced when we discussed $W_2$ geodesics in (5), $F_\alpha = T_{\frac{\alpha}{2}}$ where $T$ is a Monge map between $\mu_{\theta_n}$ and $P_x$. By the description of such geodesics $\mu_{\theta_{n+1}}$ is on the geodesic between $\mu_{\theta_n}$ and $P_x$. Now we use the uniqueness of displacement interpolation between geodesics (Villani, 2008) to obtain that this geodesic between $\mu_{\theta_n}$ and $P_x$ is a restriction of the geodesic between $\mu_{\theta_0}$ to $P_x$. As a result, the latter contains $\mu_{\theta_{n+1}}$. We conclude by induction that all the generated distributions progress toward $P_x$ on the expected $W_2$-geodesic. In Figure 4 of the experiments section, we visualize our generator following this geodesic during training.

In particular, denoting $G_\infty(z)$ the generator at convergence, we relate it to the optimal map $T$ between $\mu_{\theta_0}$ and $P_x$:

$$T \# \mu_{\theta_0} = T G_{\theta_0} \# P_z = G_\infty \# P_z \sim P_x \tag{11}$$

For more information on the speed of convergence of the generated distribution, we conduct a similar analysis in the Appendix 8.2 relying on the notion of gradient flows (Ambrosio et al., 2008). Notice that our generator produces more than the Monge map – it provides a discrete version of the $W_2$-geodesic between the distributions as we observe it during training. This is another problem of interest, see for instance the work of Seguy & Cuturi (2015).

In practice, our model is more complicated than WGAN and its extensions. First because the discriminator in W2GAN involves two functions $\phi$ and $\psi$. Second because the $c$-inequality constraint has a simpler formulation in the case of $W_1$. [5] It is thus worth asking if we could have conducted the previous analysis in the Wasserstein-1 case.

**The Wasserstein-1 case**    Our model shares a lot with GANs relying on the first Wasserstein metric, and one might wonder why we do not use e.g WGAN, WGAN-GP or WGAN-LP as they achieve state-of-the art performance in generative modelling. For the purpose of learning an optimal map, this section is meant to explain why we cannot theoretically rely on such models, at least by trying to adapt our analysis to the $W_1$ case. Experimentally, however, as WGAN and its extensions rely on a similar objective as W2GAN, the seem to be following an optimal map – at least in some cases. Let us enumerate what parts of the previous analysis does not apply to the case of $W_1$:

- The local analysis could actually be applied in a similar way. Recall that local direction given to the generated distribution by the Wasserstein GAN discriminator is $-\nabla f(.)$ where $f$ is a (non unique) Kantorovitch potential in the $W_1$ case. It is true then that there exists an associated Monge map $T$ from the current distribution toward the true distribution. This relationship is $\frac{T(x)}{\|T(x)\|} = -\nabla f(x)$ (Brenier, 1991). Hence, we can get the direction of an OT

---

[4]It is important to notice the strong assumption made on parameter's updates. We indeed assumed that each step was ideal in the sense the parametric update exactly matched the update in the space of probability measures. In practice, the limitation induced by parameters might result in deviation from the trajectory described by the Monge map. We further examine this parametric obstruction in the Appendix (8.3)

[5]This is because in the Wasserstein-1 case, $\phi$ and $\psi$ are related by $\phi = -\psi$ by a $c$-concavity argument, and that the $c$-inequality constraint is tantamount to lipschitzness of the potentials.

map thanks to the knowledge of the dual optimal variable, but not its norm. Thus, locally, the generator is updated toward the true distribution, and in the direction of the optimal map, but the magnitude of the update step is impossible to obtain.

- Going from a local to global analysis is much more difficult in the case of $W_1$. Recall that in the case of $W_2$, the main ingredient of the discussion was the uniqueness of geodesics and their nice analytic description. In the case of $W_1$, there are eventually infinitely many geodesics. A great description of this fact is available in (Bottou et al., 2018). Hence, while locally the discriminator may be one among many Kantorovitch potentials, it is hard to decide whether globally the generator follows a geodesic. Although this seems to be true as every local direction is happening on geodesics, it would remain impossible to decide which geodesic the generator is following. It would be of interest to describe different kind of $W_1$-geodesics and understand the ones that Wasserstein GANs' generators are prone to follow.

- Essentially because of the previous discussion it is also really hard to solve the gradient flow of the $W_1$ metric as we do in the Appendix 8.2 in the case of $W_2$.

In conclusion, while we might have a locally nice description of GANs relying on $W_1$, we cannot obtain a global description. Low dimensional experiments still strongly suggest that their generators might recover a Monge map at the end of training.

## 6 EXPERIMENTS

### 6.1 2-DIMENSIONAL SYNTHETIC DATA

We first consider learning optimal maps between synthetic datasets in 2-dimensional space, in which we can visualize them easily [6]. Each dataset is composed of two parts: $X$ and $Y$, each composed of 2D data, and we verify that we can learn optimal maps from $X$ to $Y$. We consider three datasets (samples are shown in Figure 1-(a)). (i) **4-Gaussians**: $X$ and $Y$ are mixtures of 4 Gaussians with equal mixture weights. The mixture centers of $Y$ is closer together than that of $X$. (ii) **Checkerboard**: $X$ and $Y$ are mixtures of uniform distributions over 2D squares, of size 4 and 5 respectively, with equal mixture weights. The mixture centers of the two distributions form an alternating "checkerboard" pattern. (iii) **2-Spirals**: $X$ and $Y$ are uniform distributions over spirals that are rotations of each other. We also show in Figure 1-(b) optimal maps between data samples obtained by the optimal assignment algorithm, which we will refer to as *Discrete-OT*. We use this as a benchmark for evaluating optimal maps obtained by various methods on these specific data samples.

We apply our proposed W2GAN model on these three datasets, where the generator takes input samples from $X$ and and maps them to $Y$. Note that we first initialize the generator as an identity function [7], in order to verify our theoretical analysis that the generator can recover globally an optimal map between its initial and target distributions. As baselines, we compare with the following methods: (i) *Barycentric-OT*: a two-step algorithm for computing optimal maps between continuous or discrete distributions introduced by Seguy et al. (2018). The algorithm is based on first computing a regularized optimal transport plan, then estimating a parametric optimal map as a barycentric projection. (ii) WGAN-GP (Gulrajani et al., 2017), and (iii) WGAN-LP (Petzka et al., 2018). Both of these GAN-based models use the $W_1$ metric as the objective of the discriminator, but differ by the form of gradient penalty on the discriminator. We similarly initialize their generators as identity functions.

We show in Figure 2 results of each of these models on the three 2D datasets. When compared to the Discrete-OT map in Figure 1-(b), we can see that our proposed W2GAN can successfully recover the Monge map in all three datasets. In comparison, Barycentric-OT performs mostly very well, but we notice a clear collapse in the mapping in the case of 4-Gaussians. WGAN-GP generally performs poorly, and we observe that training becomes especially unstable as $G(X)$ approaches $Y$ – sometimes diverging from a good solution. This happens even with more training of the discriminator and smaller generator learning rates. WGAN-LP, on the other hand, matches the performance of

---

[6]More experimental details can be found in Appnedix 8.9.1

[7]We experiment with two methods for identity initialization: reconstruction and adding a skip connection to output with very small initial weights.

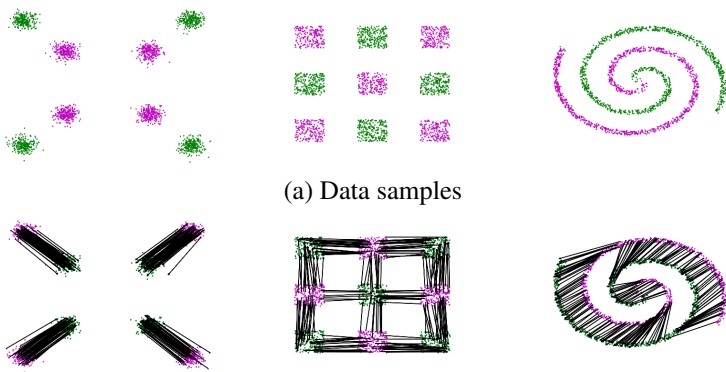

(a) Data samples

(b) Discrete-OT on data samples

Figure 1: True data samples with corresponding optimal maps (black arrows) obtained with Discrete-OT. (a) 1024 data samples of $X$ (magenta) and $Y$ (green) in 4-Gaussians, Checkerboard, 2-Spirals. (b) Optimal map computed using optimal assignment algorithm *between 200 samples from X and Y*

W2GAN in all three datasets. The fact this optimal map seems to match the Monge map of the $W_2$ case is interesting according to our analysis in 5, which indeed indicates that locally the generator is driven on some optimal maps, but cannot predict on which $W_1$ geodesic the generator will globally evolve. The experiments might suggest that WGAN and its extensions are more prone to choosing certain $W_1$ geodesics. It clearly outperforms WGAN-GP, which is unstable according to the argument of Petzka et al. (2018) that the gradient penalty in WGAN-GP can prevent the discriminator from converging.

We also verify our analysis 5 that the local direction given to the generator is $Id - \frac{\alpha}{2}\nabla\phi$, which corresponds to a small step in the global direction $Id - \nabla\phi$. This direction is also supposed to be an optimal map $T$ according to 3. We can see in Figure 3 that the approximation of the Monge map by $\phi$ is quite reasonable. This is especially interesting, given that $\phi$, as well as $\psi$, are both real functions which are not explicitly trained to recover the Monge map.

Finally, Figure 4 shows the generator's dynamics in checkerboard dataset. We see that it evolves on the optimal map.

### 6.2 HIGH-DIMENSIONAL DATA

Next, we move to the more challenging setting of learning optimal maps in high-dimensions. We first consider the task of mapping samples of a $28 \times 28$ multivariate Gaussian[8] to MNIST, which was originally proposed by Seguy et al. (2018), and compare with their method. We follow the exact experimental setting of Seguy et al. (2018). We also use the same architecture for both W2GAN generator and their Barycentric-OT network for fair comparison [9]. Figure 5 shows the generated samples (output of learned mappings) by our model and the Barycentric-OT model. Qualitatively speaking, W2GAN seems to generate much better MNIST samples, which confirms that it is a competitive method for estimating the Monge maps in high dimensions.

In this second set of experiments, we apply our model to the unsupervised domain adaptation task, which is a standard experimentation setting for evaluating large-scale OT maps (Courty et al., 2017b). We also try to follow the same experimental settings of Seguy et al. (2018), where the model learns to map across USPS and MNIST datasets. In unsupervised domain adaptation, we have labels in one dataset (e.g. USPS) and our goal is to train a classifier for the other dataset (e.g. MNIST) *without* having access to any labels in it. An optimal map is assumed to learn a mapping in the image space that preserves, as much as possible, the digit identity.

Similar to Seguy et al. (2018), we evaluate the accuracy the different models using a 1-nearest neighbour (1-NN) classifier. As a baseline, we train the classifier using source (USPS/MNIST)

---

[8]With mean and covariance matrix estimated with maximum-likelihood on MNIST training data.

[9]More experimental details can be found in Appendix 8.9.2

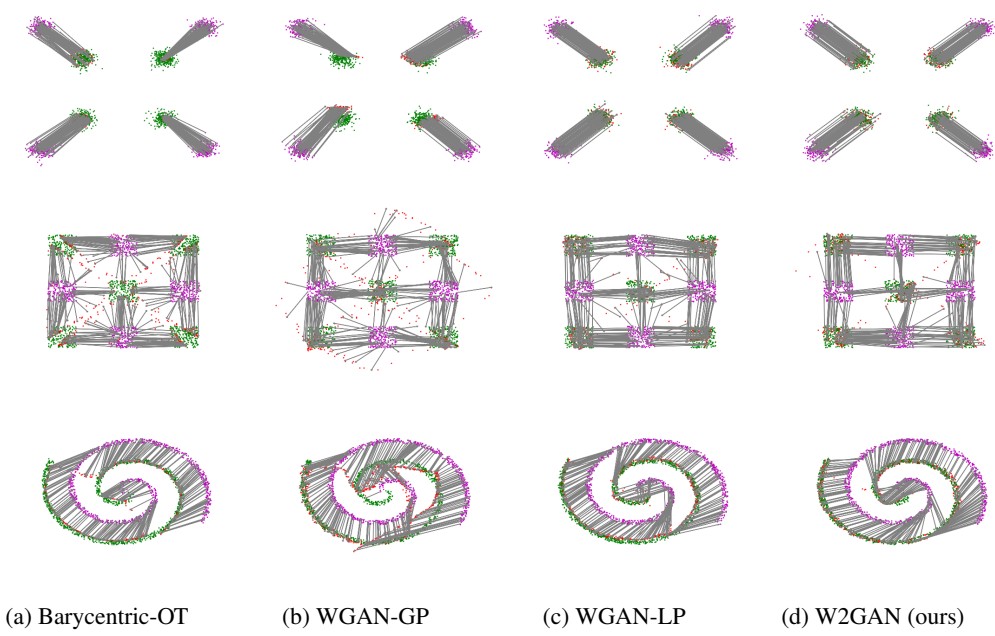

(a) Barycentric-OT      (b) WGAN-GP      (c) WGAN-LP      (d) W2GAN (ours)

Figure 2: Mappings learned in three synthetic datasets by Barycentric-OT (Seguy et al., 2018), WGAN-GP (Gulrajani et al., 2017), WGAN-LP (Petzka et al., 2018) and our proposed W2GAN. The generator maps (gray arrows) samples from $X$ (magenta) to samples $G(X)$ (red) such that it matches distribution of $Y$ (green). W2GAN and WGAN-LP can successfully recover optimal map between $X$ and $Y$ in all three datasets. The generator is initialized as identity before training, thus the analysis implies it is learning a map between $X$ and $Y$.

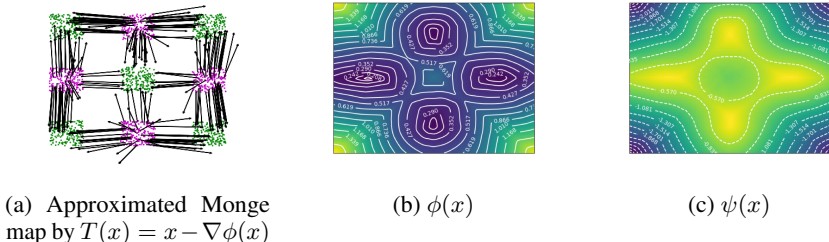

(a) Approximated Monge map by $T(x) = x - \nabla \phi(x)$        (b) $\phi(x)$        (c) $\psi(x)$

Figure 3: In W2GAN, the discriminator approximates the Monge map locally. (a) Gradient direction provided by $\phi$ to the generator. $\phi$ takes the magenta distribution as input. The arrows displayed correspond to the approximation of $T$ using the discriminator (b) Heat-map of values of $\phi$ over $\mathbb{R}^2$. (c) Heat-map of values of $\psi$ over $\mathbb{R}^2$.

images, and test it on target (MNIST/USPS) images (i.e. identity mapping). We compare this trivial mapping to the mapping obtained by both our W2GAN model and Barycentric-OT, where here the 1-NN classifier is trained on domain transferred source digits. We can see in Table 1 that our model achieves better results than Barycentric-OT. We note that our baseline is slightly lower than that of Seguy et al. (2018), possibly due to image-processing differences, and correspondingly, our implementation of Barycentric-OT also achieves slightly lower accuracy. Even so, we can achieve higher accuracy with our model compared to the result reported in their paper.

## 7    CONCLUSION AND FUTURE WORK

We believe this work offers a new perspective on GANs: a way to characterize the dynamics of the generator during training, and as a main application a way to compute a Monge map. In W2GAN, the

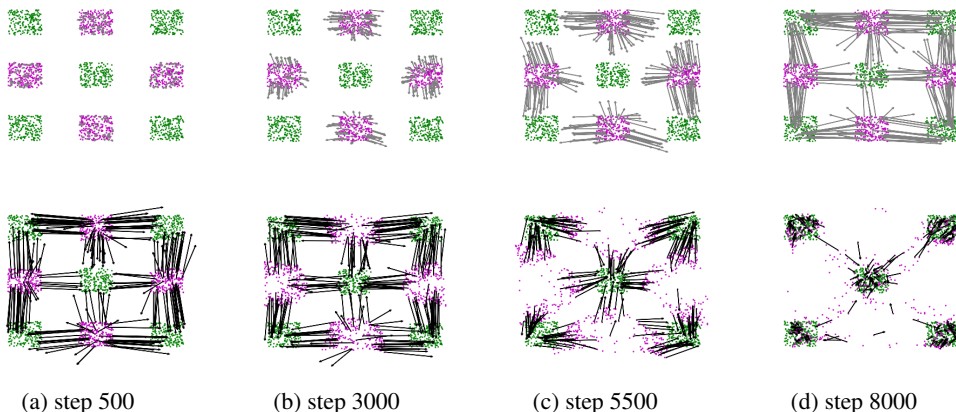

| (a) step 500 | (b) step 3000 | (c) step 5500 | (d) step 8000 |

Figure 4: Evolution of the generator (top row) and the gradient it receives from $\phi$ (bottom row) through training. The generator starts as an identity function and is trained to recover the green distribution. We can track its evolution during training with gray arrows. The gradient it receives from $\phi$ is represented by black arrows from the generated distribution $G(X)$ (magenta) toward the direction given by the discriminator's signal $G(X) - \nabla\phi(G(X))$. The generator is following the $W_2$-geodesic.

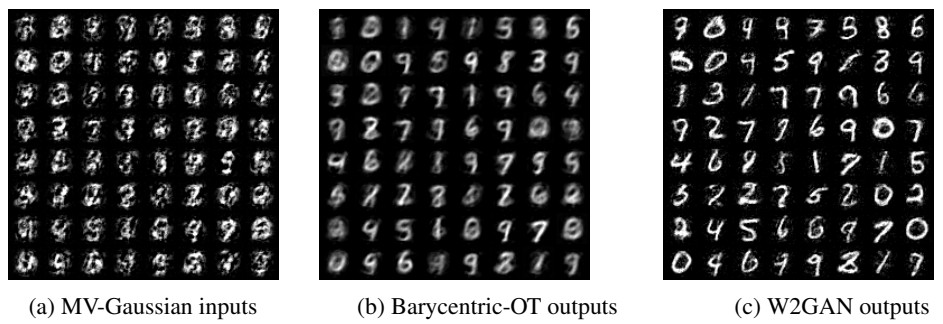

| (a) MV-Gaussian inputs | (b) Barycentric-OT outputs | (c) W2GAN outputs |

Figure 5: Comparing learned maps by W2GAN model and Barycentric-OT (Seguy et al., 2018) on the task of mapping $28 \times 28$ MV-Gaussian to MNIST. (a) Samples from MV-Gaussian with mean and covariance estimated from MNIST. (b) Corresponding samples from Barycentric-OT. (c) Corresponding samples from our W2GAN model.

|  | USPS to MNIST | MNIST to USPS |
|---|---|---|
| Source | 33.31 | 70.05 |
| Barycentric-OT (Seguy et al., 2018) | 60.50 | 77.92 |
| Barycentric-OT (our implementation) | 58.68 | 63.48 |
| W2GAN | **67.89** | **80.02** |

Table 1: Accuracy of 1-NN classifier (in %) trained using source vs. transported data.

generator recovers an optimal map between its initialized distribution and the target distribution. To establish this, we connected the generator's training procedure, and especially the signal it receives from the discriminator, with the Wasserstein-2 geodesics. In the Appendix 8.1, we raise interesting questions about our model left for future works. In particular, how to generalize this analysis of the generator dynamic during training to other GANs? How to confirm that asymptotically, when the term of regularization goes to 0, the gradient of the regularized optimal potential $\phi$ for (4) does indeed recover the relationship 3 with the optimal map $T$ solving (1)? In the appendix 8.3 we explore the down-to-earth case of update in the space of parameters. It would be also important to try relating the ideal geodesics in the space of probability measures with the actual projections in the space of parameters.

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

## 8 APPENDIX

### 8.1 REMAINING QUESTIONS AND FUTURE WORKS

In the following we detail questions raised by our model.

- A crucial relationship used in our model is (22), relating an optimal map $T$ solving the Monge problem (1) with the Kantorovitch potential $\phi$ solving the dual (2). We recall that this link is $T(x) = x - \nabla\phi(x)$ for any $x$ in the support of the initial distribution. In practice though, we get this potential by solving a regularized version of (2), for instance (4) or (27). Assuming for simplicity we deal with (4), we face a dual problem which basically removes the hard constraint defining (2) by including a penalty term $\mathcal{L}_{\text{ineq}}$ in the main objective, the emphasis of which we control with the hyperparameter $\lambda_{\text{ineq}}$. Denoting $\phi_{\lambda_{\text{ineq}}}$ the corresponding solution -which may be proven to be unique up to translation- it seems to constitute a hard challenge to prove that $\phi_{\lambda_{\text{ineq}}} \to \phi$ as $\lambda_{\text{ineq}} \to 0$. The case of the entropy penalty, that is when we define the regularization $\mathcal{L}_{\text{ineq}}(\phi, \psi) := -\mathbb{E}_{(x,y)\sim\mu\times\nu}[\exp(\frac{\phi(x)+\psi(y)-c(x,y)}{\lambda_{ineq}})]$ has been widely studied. In a discrete setting, the convergence might be obtained as a result of Cominetti & San Martín (1994), but their analysis does not straightforwardly scale to the case of absolutely continuous probability measures. What we would find more useful is to prove the convergence $\nabla\phi_{\lambda_{\text{ineq}}} \to \nabla\phi$, as it would mean that our approximation of the optimal transport map $T$ using relationship (3) is valid. Notice that the entropy regularized dual also admits a primal formulation, which solution $\pi_{\lambda_{\text{ineq}}}$ may be proven to converge in some sense to the unique transport map $\pi$ solving (19) (Carlier et al., 2017). No similar results have been found about the dual variables for the moment. Also, the case of other penalties as the $L_2$ one remain open problems.

- In a similar fashion, it would enrich our analysis to have bounds on the deviation from the ideal trajectory, the $W_2$-geodesic, when we assume some errors of approximation for the generator and the discriminator. In the same way, how far is the final generated distribution from the Monge map in this more realistic case?

- Another crucial theoretical need for strengthening our analysis would be to deepen the parametric analysis 8.3 and try to understand the trajectory of the generated distribution in the parameter space more thoroughly. Ideally, one would like to prove that this trajectory would be some projection of the $W_2$-geodesic in the space of parameters.

- In practice, what are the new possibilities enabled by the analysis of the generator's dynamic and the W2GAN model? First, it would be of interest to characterize the dynamic of GANs relying of $f$-divergences in a similar manner. In the Wasserstein-1 case, it is possible that knowing the generator is learning an optimal correspondence could be useful in a domain transfer situation, especially in high dimension where those models perform well. In particular, listing the different $W_1$-geodesics and characterizing the ones that are mainly followed by WGAN's generators could be a great insight.

- On the other hand, as the W2GAN's discriminator approximates the second Wasserstein distance, it enables tackling the challenge of computing Wasserstein barycenters of distributions (Cuturi & Doucet, 2014). In fact, it seems likely that an adversarial and parametric method would perform well in the task of generating all Wasserstein interpolations between distributions in high-dimension, a difficult problem addressed by Xie et al. (2018) for instance.

### 8.2 CONTINUOUS ANALYSIS OF THE GENERATOR'S EVOLUTION

We here strengthen the argument that during training, the generated distribution $\mu_\theta$ is following the "line" defined the optimal transport map between its initialization $\mu_{\theta_0}$ and the target distribution $P_x$. As in the previous discussion, we assume the discriminator $(\phi, \psi)$ to compute exactly the squared second Wasserstein distance $W_2^2(\mu_\theta, P_x)$, i.e. we suppose it is trained infinitely many times at

each update of the generator, and we forget about the bias induced by the c-inequality constraint being encoded in the objective as a penalty (informally, we assume $\lambda_{\text{ineq}} = 0$). We again look at the evolution of the generated distribution in the case of ideal updates in the space of probability measures. The difference in this case is that we consider $\alpha \to 0$, thus writing $G(t, Z)$ as a time-dependent random variable where G is in the space of $L_2$ functions and $t$ is a fictive time variable. In this case we may equivalently work on the corresponding generated probability distribution of interest, which we denote as $\mu_t := \mu_{\theta_t}$. Then our generated distribution evolves according to the gradient flow

$$\dot{\mu}_t = -\nabla W_2^2(\mu_t, P_x). \tag{12}$$

One would need to introduce the definition of gradient flow in the space of probability measures, in particular the notion of real functions' gradient with respect to probability measures and velocities of time-dependent measures in order to fully express the meaning of the above. We refer to Ambrosio et al. (2008) for a comprehensive overview. We can refine (12) thanks to Ambrosio et al. (2004) and obtain:

$$\dot{\mu}_t = T_t - Id \tag{13}$$

where $T_t$ is the unique optimal transport map, Solving (1) between $\mu_t$ and $P_x$. According to the gradient flow (13), locally, we recover that the generated distribution evolves towards $P_x$ by following the Wasserstein-2 geodesic. In fact, this is a global behaviour, from Ambrosio et al. (2008):

**Theorem 1.** *Denote $T$ the optimal transport map between $\mu_0$ and $P_x$. Then we have that the gradient flow solving* (12) *is uniquely determined:*

$$\mu_t = [e^{-t}Id + (1 - e^{-t})T]\#\mu_0 \tag{14}$$

Hence a consequence (also from Ambrosio et al. (2008)) is that the generated distribution evolves exponentially fast towards $P_x$:

**Corollary 1.**

$$\forall t \geqslant 0, W_2^2(\mu_t, P_X) = e^{-2t}W_2^2(\mu_0, P_x) \tag{15}$$

The evolution in (14) suggests that the generated distribution follows the Wasserstein-2 geodesic between $\mu_0$ and $P_x$. That means the training dynamic of the generator "draws" the optimal transport map between $\mu_0$ and $P_x$. At the end of training, the generator $G(\infty, .)$ provides a certain optimal transport map. For each $z$, the 'arrows' joining $G(0, z)$ and $G(\infty, z)$ together constitute the optimal map. Figure 7 in the Appendix helps visualizing this analysis.

## 8.3 THE PARAMETRIC CASE

We now turn to the more realistic case of a parametrized generator $G_\theta(z)$, i.e we will analyse the update in the space of parameters. Lui et al. (2017) already made the conjecture that a generator trained with the second Wasserstein distance would have its parameters updated towards the direction of an optimal transport. Let us prove their statement in the case of our model. In the following, $J_f(u)$ denotes the Jacobian of a function $f$ at point $u$. Let $G_\theta(z)$ be our parametrized generator, which we assume to admit derivatives w.r.t both the parameters $\theta$ and $z$. Recall the notation $\mu_\theta := \mathbb{P}_{G_\theta(z)} = G_\theta\#\mathbb{P}_z$ where $\mathbb{P}_z$ is the measure known beforehand on the latent variable $z$. We consider the context of alternating gradient descent with learning rate $\alpha > 0$ for the generator. We naturally associate the fictive time variable t, so that we consider the discrete update equation:

$$\theta_{t+1} = \theta_t - \alpha \frac{\partial W_2^2(\mu_{\theta_t}, P_x)}{\partial \theta} \tag{16}$$

**Proposition 1.** At each generator update, we assume the discriminator $\phi, \psi$ to achieve $\sup_{\phi,\psi} V^*(\phi, \psi, \mu_\theta, P_x) = W_2^2(\mu_\theta, P_x)$ where $V^*(\phi, \psi, \mu_\theta, P_x) := \int_{\mathbb{R}^d} \phi(x)dP(x) + \int_{\mathbb{R}^d} \psi(x)dP_x$ is the value function of the dual Kantorovitch problem (2). Then (16) admits a tractable form as:

$$\frac{\partial W_2^2}{\partial \theta} = \mathbb{E}_{z\sim P_z}(J_{\phi(G_\theta(z))}^T(\theta)) = \mathbb{E}_{z\sim P_z}(\nabla_\theta \phi(G_\theta(z)))$$

Moreover, the gradient ascent dynamics of the generator are linked to the optimal transport map $T$ solving (1) between $\mu_\theta$ and $P_x$ as:

$$\forall z, G(\theta_{t+1}, z) = G(\theta_t, z) + \alpha J_{G(.,z)}^T(\theta_t) \times \mathbb{E}_{z\sim P_z}\left(J_{G(.,z)}^T(\theta_t) \times [T(G(\theta_t, z)) - G(\theta_t, z)]\right) + o(\alpha) \tag{17}$$

In the above, we use the notation $G(\theta, z) := G_\theta(z)$ to clarify where the derivatives are taken. Equation (17) is close to the one conjectured in Lui et al. (2017). Following their approach, for the sake of clarity, we consider the case where the latent variable $z$ is a constant. Then we read (17) as

$$G(\theta_{t+1}, z) = G(\theta_t, z) + \alpha J_{G(.,z)}(\theta_t) J_{G(.,z)}^T(\theta_t) \times [T(G(\theta_t, z)) - G(\theta_t, z)]$$

As similarly observed in Lui et al. (2017), if we ignore the possibly significant effect of $J_{G(.,z)}(\theta_t) J_{G(.,z)}^T(\theta_t)$, the term $T(G(\theta_t, z)) - G(\theta_t, z)$ enforces G to be updated on the optimal transport plan joining $\mu_\theta$ and $P_x$.

We want to highlight the strong sensitivity of the model to any error of approximation. The dynamics of the generator is easily perturbed by biased updates. Let us denote the generated distribution at time $t = k$ by $\mu_k$. Let us also introduce $g_k$ as the Wasserstein geodesics between $\mu_k$ and $P_x$. Imagine, for instance, that there exists $k > 0$, such that $\mu_{k+1}$ is not on the optimal path between $\mu_k$ and $P_x$. Then, from now on, the generator will evolve ideally on the Optimal Transport map joining $\mu_{k+1}$ and $P_x$, and thus will no longer evolve in the previous geodesic. More succinctly, $g_{k+1} \neq g_k$. In some sense, the learning dynamics of the generator is a Markov process in that it "forgets" the previous updates. A wrong update leads to a generator that no longer evolves exactly on the optimal map between the initial $\mu_0$ and the target $P_x$. It would certainly be of interest to have a mathematical framework that accounts for this variance in the evolution of the generator, but this is beyond the scope of this paper.

## 8.4 RESULTS IN OPTIMAL TRANSPORT THEORY

We develop a bit more the materials of the background section, introducing the same notions with more details in the same order.

**Monge Problem** Optimal Transport (OT) theory (Villani, 2008; Santambrogio, 2010; Ambrosio & Gigli, 2013) introduces a natural quantity to distinguish two probability measures. Given two probability measures $\mu$ and $\nu$ on the euclidean space $\mathbb{R}^d$, the original *Monge problem* is to find a *map $T$* that "transports" the $\mu$ distribution on $\nu$, and minimizes the cost of the transport, which is point-wise defined by a fixed cost function $c : \mathbb{R}^d \times \mathbb{R}^d \to \mathbb{R}$. The value $c(x, y)$ can be seen as the cost for transporting a unit from $x$ to $y$. The problem is summarized by:

$$\inf_{T \in \mathcal{A}_T} \int_{\mathbb{R}^d} c(x, T(x)) d\mu(x) \tag{18}$$

where $\mathcal{A}_T$ is the set of all maps from $\mathbb{R}^d$ to $\mathbb{R}^d$ that respect the marginal $\nu$, more formally written as $T_\# \mu = \nu$. The map $T_\# \mu$ is called the *push-forward measure* of $\mu$ and is defined as $T_\# \mu(A) = \mu(T^{-1}(A))$ for any Borel set $A \in \mathbb{R}^d$. Figure 6 in the Appendix provides an illustration of the shape of an optimal transport map $T$ solving (1).

**Kantorovitch relaxation** Unfortunately, problem (1) often does not admit a solution as $\mathcal{A}_T$ might be empty [10]. To circumvent this issue, one considers the so-called *Kantorovitch relaxation* of this problem:

$$V_c(\mu, \nu) := \inf_{\pi \in \mathcal{A}(\mu,\nu)} \int_{x,y \in \mathbb{R}^d} c(x, y) d\pi(x, y) \tag{19}$$

where $\mathcal{A}(\mu, \nu)$ is the set of joint distributions whose first and second marginals are equal to $\mu$ and $\nu$ respectively. The joint $\pi$ is called the *transport plan* between $\mu$ and $\nu$. If $\pi$ is deterministic — specifically, for any $x$, there is a unique $y$ such that $\pi(x, y) > 0$ — then it is also a transport map as defined in the Monge problem. It suffices to define such a map $T(x)$ by the only $y$ respecting $\pi(x, y) > 0$. However, we could instead consider $\pi$ to be a "one-to-many" transport plan: for each $x$, there might be several $y$ such that $\pi(x, y) > 0$. While Monge's development had the problem of non-existence of the transport map, the Kantorovitch relaxation $\mathcal{A}(\mu, \nu)$ is never empty. In particular, it always contains the independent joint distribution $\mu \times \nu$. In many situations, under mild assumptions on the cost function $c$ (semi-lower continuity and bounded from below), there always

---

[10]For instance, let us take $\mu = \delta_0$ a dirac at 0 and $\nu := \frac{1}{2}(\delta_{-1} + \delta_1)$ a weighted sum of diracs, both on real line. One can see that any map T would have to send 0 on either $-1$ or 1, hence the constraint $T\#\mu = \nu$ cannot hold

exists a minimizer of (19), so that the infimum might be replaced by a minimum (Ambrosio & Gigli, 2013). We distinguish between a *transport map* $T$ achieving the minimum in (1) and a *transport plan* $\pi$ minimizing (19). Note that for most applications, we are more interested in obtaining an approximation of the optimal transport map than a transport plan.

**Dual of Kantorovitch** One could also work with the dual of problem (19), which is proven to lead to the same value $V_c(\mu, \nu)$. We write this dual as:

$$\sup_{\phi, \psi \in \mathcal{A}^*(\mu, \nu)} \int_{x \in \mathcal{X}} \phi(x) d\mu(x) + \int_{y \in \mathcal{Y}} \psi(y) d\nu(y) \tag{20}$$
$$\mathcal{A}^*(\mu, \nu) := \{(\phi, \psi) : \mathbb{R}^d \to \mathbb{R} / \ \forall x, y \in \mathcal{X} \times \mathcal{Y}, \phi(x) + \psi(y) \leqslant c(x, y)\}$$

We will also denote as $V^*(\phi, \psi)$ the value $\int_{x \in \mathcal{X}} \phi(x) d\mu(x) + \int_{y \in \mathcal{Y}} \psi(y) d\nu(y)$. A pair $(\phi, \psi)$ maximizing $V^*$ are called *Kantorovitch potentials*. We will refer to the constraint in the definition of (2) which forces the sum of the potentials $\phi$ and $\psi$ to be upper-bounded by the cost as the *c-inequality constraint* or simply the *inequality constraint*.

**Wasserstein distances** When the cost $c$ is a distance, $V_c(\mu, \nu)$ exactly matches the famous $W_1(\mu, \nu)$, which is called the first Wasserstein distance between $\mu$ and $\nu$. When the cost $c$ is a distance to a power of some positive integer $p$, $V_c(\mu, \nu)^{1/p}$ is denoted $W_p(\mu, \nu)$ and is commonly called the $p^{\text{th}}$ Wasserstein distance. An important result is that these $W_p$ distances are actually respecting the axioms of a distance over distributions (Villani, 2008). For the sake of simplicity, we will abusively extend the term "Wasserstein distance" to any $V_c$, for any cost function c, even though not all choices of $c$ lead to $V_c$ being a metric over probability distributions.

**Computing Wasserstein distances** The discriminator of the Wasserstein GAN (WGAN) (Arjovsky et al., 2017) computes the dual formulation (2), with $\nu$ being the "true" distribution $\mathbf{X}$ (defined by training examples) and $\mu$ being the "fake" distribution $\mathbf{G}(\mathcal{Z})$ (defined by samples drawn from the generator). In practice, the inequality constraint on potentials $\phi, \psi$ may be enforced by the addition of a constraint-violation penalty term to the objective (Gulrajani et al., 2017). Although introduced for other reasons, Cuturi (2013) proposed an efficient penalized version of (19) whose resulting optimization problem is called the entropic-regularized optimal transport problem. We shall only provide its dual form:

$$\sup_{\phi, \psi} V^*(\phi, \psi) + \lambda_{\text{ineq}} \mathcal{L}_{\text{ineq}}(\phi, \psi) \tag{21}$$

where $\mathcal{L}_{\text{ineq}}$ penalizes $(\phi, \psi)$ when they violate the inequality constraint:

$$\mathcal{L}_{\text{ineq}}(\phi, \psi) := -\mathbb{E}_{(x,y) \sim \mu \times \nu} \left[ \exp \left( \frac{\phi(x) + \psi(y) - c(x, y)}{\lambda_{\text{ineq}}} \right) \right]$$

Other penalties were theoretically explored by Blondel et al. (2017), including the widely used $L_2$-penalty (Seguy et al., 2018):

$$\mathcal{L}_{\text{ineq}}(\phi, \psi) := -\mathbb{E}_{(x,y) \sim \mu \times \nu} \left[ (\phi(x) + \psi(y) - c(x, y))_+^2 \right]$$

where $(.)_+ := \max(0, .)$. Objective (21) was shown to asymptotically recover the value of (2) when $\lambda_{\text{ineq}} \to 0$ (Carlier et al., 2017).

Due to its suitability for training parametric models such as neural network, we exploit this optimization objective to compute Wasserstein distances $W_p$. As it is provable (Ambrosio & Gigli, 2013) that the 'differential' of $W_p(., \nu)$ with respect to its first variable is $-\phi$ (where $\phi$ is a Kantorovitch potential), we may conclude that a GAN framework where the discriminator accurately computes (2) through (21) provides a means of training a generative model. The details of such a framework are developed in the next section.

**Links between the Kantorovitch potentials and the Monge map** We end this section by exploring the relationship between optimal potentials and optimal transport plans and maps. The goal is to recover an approximation of the solution of (1), namely an optimal transport map, given optimal solutions of (2) (i.e. Kantorovitch potentials). Getting an approximate transport plan might be

done by considering a specific relationship that holds between the solutions of the dual and the primal in regularized versions of the optimal transport (Cuturi, 2013; Xie et al., 2018). As for an optimal transport map, Seguy et al. (2018) provides a two-step procedure, that is, they first compute a regularized transport plan, and then calculate an approximate transport map by doing a barycentric projection of the plan. In our case, we begin by the observation that there is no distinction between an optimal plan and its corresponding map in the case of the Wasserstein distances $W_p$ with $p > 1$. In addition, the primal-dual relationship is straightforward. Getting an optimal transport map solving (1) is a by-product of computing Kantorovitch potentials by solving (2). We summarize this famous statement (Brenier, 1991):

**Proposition 2.** *Fix $p \geqslant 2$ and the cost $c(x, y) := \frac{\|x-y\|^p}{p}$. Then there is one unique optimal transport plan $\pi$ solving (19). It is deterministic, that is it corresponds to an optimal transport map $T$ solving (1): $\pi = (Id \times T)$. Also, the Kantorovitch potentials $\phi, \psi$ are unique up to a translation and the following relation holds:*

$$\forall x \in supp(\mu), \ \ T(x) = x - \|\nabla\phi(x)\|^{\frac{1}{p-1}-1}\nabla\phi(x) \tag{22}$$

Finally, we recall a useful result for the converse question: given a function $T$ from $R^d$ to itself and a probability $\mu$, is $T$ a Monge map between $\mu$ and $T\#\mu$? In the case of the second Wasserstein distance, Brenier's polarization theorem (Brenier, 1991) says this is true if $T$ is the gradient of a strictly convex function. If $T$ may be written $x - \nabla\phi(x)$ for some real function $\phi$ one can rephrase this condition as $\frac{\|x\|_2^2}{2} - \phi(x)$ is strictly convex. Such a $\phi$ is said to be c-concave. In particular, any Kantorovitch potential $\phi$ is c-concave.

**Geodesics in the space of probability measures**   Recall that $W_2$ is a metric over distributions and thus makes the space over probability measures a metric space. Studying properties of this space starts with the analysis of its geodesics. That is, for any given distributions $\mu$ and $\nu$, we look for constant speed paths described by $\mu_t$ such that $\mu_0 = \mu$, $\mu_1 = \nu$ and

$$\forall 0 \leqslant s \leqslant t \leqslant 1, W_2(\mu_s, \mu_t) = (t - s)W_2(\mu, \nu)$$

It happens that such geodesics are unique (Villani, 2008). Given a Monge map $T$ between $\mu$ and $\nu$, the only constant speed geodesic is given by $\mu_t := T_t\#\mu$ where $T_t = (1 - t)Id + tT$. It is a remarkable fact, compared with the case of the first Wasserstein distance, where the geodesics are eventually in infinite number. This is one obstruction to conduct directly our analysis in this case, and this is one reason we appeal to $W_2$ instead of $W_1$.

By 3, we can write $T(x) = x - \nabla\phi(x)$ for any Kantorovitch potential $\phi$ solving (2). Hence $T_t(x) = x - t\nabla\phi(x)$. In particular, this guarantees that $t\phi(x)$ is a Kantorovitch potential solving (2) for $\mu$ and $\mu_t = T_t\#\mu$.

## 8.5   PROOFS

*Proof of proposition 2.* This result is a particular case of a well-known correspondence between Kantorovitch potentials and optimal transport map. In fact, when the cost $c$ is such that $c(x, y) = h(x - y)$ and $h$ is strictly convex, one has the existence and uniqueness of an optimal Monge map $T$ solving (1) and a specific relationship with the Kantorovith potential $\phi, \psi$ solving (2) Villani (2008); Santambrogio (2017):

$$T(x) = x - (\nabla h)^{-1}(\nabla\phi(x))$$

Fix $p \geqslant 2$ and consider the case of the p-Wasserstein distance. Then $c(x, y) = h(x - y)$ where $h(x) := \frac{1}{p}\|x\|^p$. A norm is always convex by triangle inequality, and any $x \to x^p$ is strictly convex and increasing on $\mathbf{R}_+$, so the previous result provides the uniqueness of the optimal transport map $T$. It only remains to invert the gradient of h. A quick calculation gives:

$$\forall y \in Im\nabla h, \nabla h^{-1}(y) = \|y\|^{\frac{1}{p-1}-1}y$$

as we supposed the norm to be the euclidean one. Plugging this into the first expression, we obtain the desired result.   $\square$

*Proof of proposition 1.* Proving the first part of proposition 2 is exactly similar as the way it is done for the Wasserstein-1 case in Arjovsky et al. (2017). We recall that the main ingredients of the proof is first to convey an envelop theorem to obtain that

$$\nabla_\theta W_2^2(\mu_{\theta_t}, P_x) = \nabla_\theta \mathbb{E}_{z \sim P_z}[\phi(G(\theta, z))]$$

and second to use a dominated convergence argument to invert the expectation and the gradient operator in the right-hand side of the above.

For the second part of the proof, we rewrite the gradient ascent equation (16) for the generator parameter $\theta$:

$$\theta_{t+1} = \theta_t - \alpha \nabla_\theta W_2^2(\mu_{\theta_t}, P_x) = \theta_t - \alpha \mathbb{E}_{z \sim P_z}[\nabla_\theta \phi(G(\theta_t, z))] = \theta_t - \alpha \mathbb{E}_{z \sim P_z}[J_{\phi(G(\theta_t, z))}^T(\theta_t)]$$

$$= \theta_t - \alpha \mathbb{E}_{z \sim P_z}[((\nabla \phi(G(\theta_t, z)))^T J_{G(.,z)}(\theta_t))^T] = \theta_t - \alpha \mathbb{E}_{z \sim P_z}[(J_{G(.,z)}(\theta_t)^T \nabla \phi(G(\theta_t, z))]$$

Therefore, for a fix latent variable $z_0$,

$$G(\theta_{t+1}, z_0) = G(\theta_t - \alpha \mathbb{E}_{z \sim P_z}[J_{G(.,z)}(\theta_t)^T \nabla \phi(G(\theta_t, z))], z_0)$$

Then by first order Taylor expansion:

$$G(\theta_{t+1}, z_0) = G(\theta_t, z_0) - \alpha J_{G(.,z_0)}(\theta_t) \mathbb{E}_{z \sim P_z}[J_{G(.,z)}(\theta_t)^T \nabla \phi(G(\theta_t, z)] + o(\alpha)$$

We conclude using the hypothesis that $\phi$ is maximizing the dual of Kantorovitch and is related to an optimal map T through (3). ☐

## 8.6 EXPLOITING A PRIMAL-DUAL RELATIONSHIP AND INTERPOLATING THE CONSTRAINT TO IMPROVE THE MODEL

In the following, we explain two important modifications of the objective functions for the discriminator inspired by optimal transport theory. In order to unify notations, we consider the two random variables $X$ and $Y$ to be discriminated by $\phi, \psi$. In the GAN context, one may think of $X$ as the generated distribution $G(Z)$ and $Y$ as the true distributions. In both cases the discriminator objective is divided into two parts. First, a main objective $\mathcal{L}_{\text{OT}}(\phi, \psi, X, Y) := \mathbb{E}(\phi(X) + \psi(Y))$ which corresponds to the dual of the Kantorovitch problem. Second, the inequality constraint $\mathcal{L}_{\text{ineq}}(\phi, \psi, X, Y) := -\lambda_{\text{ineq}} \mathbb{E}[(\phi(X) + \psi(Y) - \frac{\|X-Y\|_2^2}{2})_+^2]$. One could prefer an other choice of regularization such as the entropy penalty. Hence the overall objective for the discriminator is:

$$\sup_{\phi, \psi} \mathcal{L}_{\text{OT}}(\phi, \psi, X, Y) + \mathcal{L}_{\text{ineq}}(\phi, \psi, X, Y)$$

The first idea is to take advantage of relation (3), which we know to be true between an optimal potential $\phi$ and an optimal map $T$. In fact, optimal transport theory asserts that the inequality constraint should be exactly saturated where there is some transport (Villani, 2008):

**Theorem 2.** *Consider any lower-semicontinuous cost function $c$ and a optimal transport plan for* (19)*, and Kantorovitch potentials $\phi, \psi$ for* (2)*. Then,*

$$\forall x, y, (x, y) \in supp(\pi) \implies \phi(x) + \psi(y) = c(x, y) \tag{23}$$

In our case, that is when the cost is a $p^{th}$ power of the euclidean distance, we know from Proposition 1 than an optimal transport plan is actually an optimal transport map $T$, and we dispose of a relationship with the corresponding Kantorovitch potential $\phi$. Hence an immediate consequence of the above for the case of the square of the euclidean distance is:

**Corollary 2.** *For the Kantorovitch problem* (1) *related to $c(x, y) := \frac{\|x-y\|_2^2}{2}$, that is the computation of the second Wasserstein distance, given Kantorovitch potentials $\phi, \psi$:*

$$\forall x \in supp(\mu), \phi(x) + \psi(x - \nabla\phi(x)) = \frac{\|\nabla\phi(x)\|_2^2}{2} \tag{24}$$

Thus we suggest enforcing our discriminator to abide by relation (24). Notice that the previous corollary admits an exact symmetric relationship involving the gradient of $\psi$. This is done by adding the following penalty during training:

$$\mathcal{L}_{\text{eq}}(\phi, \psi, X, Y) := -\lambda_{\text{eq}} \left[ \left( \phi(X) + \psi(X - \nabla\phi(X)) - \frac{\|\nabla\phi(X)\|_2^2}{2} \right)^2 \right. \tag{25}$$

$$\left. + \left( \phi(Y - \nabla\psi(Y)) + \psi(Y) - \frac{\|\nabla\psi(Y)\|_2^2}{2} \right)^2 \right] \tag{26}$$

We call it the c-equality penalty term as it tries to enforce the dual functions to saturate to c-inequality constraint at the right locations. Hence the overall objective of the discriminator is:

$$\sup_{\phi,\psi} \mathcal{L}_{\text{OT}}(\phi, \psi, X, Y) + \mathcal{L}_{\text{ineq}}(\phi, \psi, X, Y) + \mathcal{L}_{\text{eq}}(\phi, \psi, X, Y) \tag{27}$$

**Remark 1.** It is interesting to bridge this objective with the one used in GAN relying on the first Wasserstein distance. In WGAN-GP, the discriminator is asked to have gradient exactly equal to $1$. Translating that into the optimal transport theory, it is tantamount to have potentials $\phi, \psi$ saturating the c-inequality. That is, this gradient penalty term is similar to our $\mathcal{L}_{\text{eq}}$. Importantly though, Petzka et al. (2018) raised the fact that it is not a valid practice according to theory to ask the gradient norm of the discriminator to be 1 everywhere. Instead, WGAN-LP is a model where the gradient's norm is enforced to be less than 1, which is then translated into optimal transport perspective as potentials $\phi, \psi$ respecting the c-inequality. Their penalty term is thus similar to our $\mathcal{L}_{\text{ineq}}$. Our model takes advantage of the two forms. The difference is that our $\mathcal{L}_{\text{eq}}$ is completely justified -at least because it enforces a relationship which is true at optimum- thanks to the theory existing in the $W_2$ case.

A second modification concerns the way we encode the penalty enforcing the inequality constraint to be respected by $\phi, \psi$. In fact, we modify $\mathcal{L}_{\text{ineq}}$ in order to bring it closer to the theory. Looking at the definition of $\mathcal{A}^*(\mu, \nu)$, the set of constraint of the dual of Kantorovitch problem (2), we see that the c-inequality constraint should be respected point wise everywhere in the ambient space where the distributions are defined. Instead, the entropy regularization or the $L^2$ one only enforce it for pairs $x, y$ that live in the supports of the two distributions. We want to reduce this bias by somehow enforcing the inequality on a broader set. It is sufficient that the potentials respect the inequality constraint point wise on a convex compact set $\Omega$ containing the support of the two distributions. Hence we suggest enforcing it on the convex envelop of the two supports: $\Omega := \text{Conv}(\text{supp}(\mu) \bigcup \text{supp}(\nu))$. To do so, we define two i.i.d random variables $\tilde{X}$ and $\tilde{Y}$ which follow the same law as $\epsilon X + (1 - \epsilon)Y$ where $\epsilon \sim \mathcal{U}([0, 1])$. Hence the overall objective for $\phi, \psi$ is:

$$\sup_{\phi,\psi} \mathcal{L}_{\text{OT}}(\phi, \psi, X, Y) + \mathcal{L}_{\text{ineq}}(\phi, \psi, X, Y) + \mathcal{L}_{\text{ineq}}(\phi, \psi, \tilde{X}, \tilde{Y}) + \mathcal{L}_{\text{eq}}(\phi, \psi, X, Y) \tag{28}$$

**Remark 2.** This interpolation idea has already been used in GANs relying on the first Wasserstein distance, such as WGAN-GP and WGAN-LP in their gradient penalty. This practice was more motivated by better results. Here we provided a theoretical argument in favor of such practice. On the other hand, models trying to broaden GANs to higher order Wasserstein distances and/or to compute optimal transport map (Sanjabi et al., 2018; Seguy et al., 2018; Salimans et al., 2018) in a similar manner only enforced the constraint on the support of the distributions.

**Remark 3.** We motivated the definition of $\tilde{X} := \epsilon X + (1 - \epsilon)Y$ so that it basically takes value in the convex envelop of X and Y. But in all generality, convex interpolation between two points cannot recover all $\text{Conv}(\text{supp}(\mu) \bigcup \text{supp}(\nu))$. By Carathodory theorem, one would need to interpolate at most between $d + 1$ points to recover all of it. Although not impossible at all, we wanted to keep the model simple and not depend on the dimension of the latent space $\mathbb{R}^d$.

## 8.7 Parameterizing $\phi$ and $\psi$ for stability

An intuitive way of parameterizing $\phi$ and $\psi$ is to simply replace $\phi$ and $\psi$ with two neural networks of the same architecture. A potential downside we found practically is that this parameterization tends to be unstable. An alternative reparameterization is to replace $\psi(Y)$ with $-\phi(Y) + \epsilon(Y)$ where both $\phi$ and $\epsilon$ are neural networks. This reparameterization and the property that

$$\phi(Y) + \psi(Y) \leq 0$$

---

**Algorithm 1** W2GAN with $D = (\phi, \psi)$ and $L^2$-regularization.

---

**Require:** latent space Z and true distribution X from which we have an available sample procedure.
**Require:** $\lambda_{\mathrm{eq}}, \lambda_{\mathrm{ineq}}, \lambda_\epsilon, n_{\mathrm{critic}}, bD, p$.
**Require:** Initial critic parameters $w_0$,
  initial generator parameters $\theta_0$.
  **while** $\theta$ has not converged **do**
    Initialize generator and discriminator losses $\mathcal{L}_D, \mathcal{L}_G$ to 0.
    **for** $t = 1, ..., n_{\mathrm{critic}}$ **do**
      **for** $i = 1, ..., bD$ **do**
        Sample real data $x, x' \sim \mathbf{P}_X$, latent variable $y, y' \sim \mathbf{P}_{G(Z)}$ and $\epsilon_1, \epsilon_2 \sim \mathcal{U}([0, 1])$.
        Update $\mathcal{L}_D$ according to 28 and optionally 29.
      **end for**
      Update $(\phi, \psi)$ with respect to $\mathcal{L}_D$.
    **end for**
    $\mathcal{L}_G \leftarrow \mathcal{L}_{\mathrm{OT}}$.
    Update G with respect to $\mathcal{L}_G$.
  **end while**

---

yields an additional regularizer

$$\mathcal{L}_\epsilon(\phi, \psi, X, Y) = -\lambda_\epsilon \left[ \epsilon(Y)_+^2 \right] \tag{29}$$

which we found particularly useful in the high dimensional setting.

## 8.8 ALGORITHMS

In algorithm 1, one can choose to use the equality constraint as an additional constraint for the discriminator. One can also use the interpolation method, or any method of sampling in 8.6.

## 8.9 IMPLEMENTATION DETAILS AND ARCHITECTURE

Below we describe the implementation details of our experiments.

### 8.9.1 2D SETTING

For the 2D synthetic data experiments, the learning rate used for Barycentric-OT is 0.005 (although we did not notice that the learning rate influenced the solution quality significantly). For the GAN experiments, learning rates were chosen from the set $\{0.00001, 0.0005, 0.00005, 0.00001\}$. For the Wasserstein-based GANs, the number of discriminator updates per generator update is chosen from the set $\{5, 10, 20\}$. This is set to 1 in the Jensen-Shannon based GAN by default. $\lambda_{\mathrm{gp}}$ for both WGAN-LP and WGAN-LP is set to 10, following their conventions. For W2-OT and W2GAN, we set $\lambda_{\mathrm{eq}} = \lambda_{\mathrm{ineq}} = 200$. To enforce that $G_{\theta_0}(z) = z$, we parameterize G by $G(z) = H(z) + z$, where $H(z)$ is initialized to be close to 0. $H$ is parameterized by 4 fully connected hidden layers of size 128, with ReLU activations and batch norm in between the layers, and 1 fully connected final layer. $\phi$ and $\psi$ are each parameterized by 2 fully connected layers with ReLU activations in between, and 1 fully connected final layer.

### 8.9.2 MULTIVARIATE GAUSSIAN TO MNIST SETTING

In this experiment, MNIST images are kept at their original size of $28 \times 28$ and pixel values are re-scaled to be in $[-1, 1]$. For Barycentric-OT, we use the same architecture as (Seguy et al., 2018). We searched over $\lambda \in \{0.01, 0.05, 0.1, 0.5, 1, 2, 5\}$, and use the ADAM optimizer with learning rate= 0.0002 and $\beta_1 = 0.5$, $\beta_2 = 0.999$ for both the dual variables and the mapping. We run the experiment with a batch size of 64 for $200,000$ iterations for each phase.

For W2GAN, to enforce that $G_{\theta_0}(z)$ is close to identity, and that $G(z) \in [-1, 1]$ we reparameterize G by $G(z) = 2 \cdot H(z) + z_{\mathrm{clip}}$ where $z_{\mathrm{clip}}$ is $z$ clipped to be in $[-1, 1]$ and tanh is used as the

final activation layer of $H$. We use the same architecture as barycentric-OT with the addition of BatchNorm in between the layers of the generator. Specifically, the architecture for our model is FC(28*28 → 1024)-BN-RELU-FC(1024 → 1024)-BN-RELU-FC(1024 → 28*28)-Tanh for $H$ in the generator, and FC(28*28 → 1024)-RELU-FC(1024 → 1024)-RELU-FC(1024 → 1) for both $\phi$ and $\epsilon$ in the discriminator. We note that in the high dimensional setting, better training stability and image quality is achieved by using both $\mathcal{L}_{eq}$ and $\mathcal{L}_\epsilon$ which complement $\mathcal{L}_{ineq}$ in enforcing the constraint. We set $\lambda_{ineq} = \lambda_{eq} = \lambda_\epsilon = 10$ and use the ADAM optimizer with learning rate = 0.0001 and $\beta_1 = 0.5$, $\beta_2 = 0.999$ for both the generator and the discriminator. We ran the experiment for $100,000$ iterations with a batch size of 64.

### 8.9.3 UNSUPERVISED DOMAIN ADAPTATION SETTING

The USPS dataset consists of $16 \times 16$ grayscale images of digits, with significantly less training and testing data (7291 train and 2007 test images). The MNIST digits are rescaled to $16 \times 16$ to match the USPS digits, and the grayscale pixels in both datasets are scaled to be in $[-1, 1]$. For this set of experiments, we use the same architecture as (Seguy et al., 2018) for Barycentric-OT, and choose the best model between using entropy regularization and L2 regularization and $\lambda \in \{0.01, 0.05, 0.1, 0.5, 1, 2, 5\}$. We use the ADAM optimizer with learning rate= 0.0002 and $\beta_1 = 0.5$, $\beta_2 = 0.999$ for both the dual variables and the mapping. We ran the experiment for $20,000$ iterations for each phase with a batch size of 1024.

For W2GAN, to ensure that $G_{\theta_0}(z)$ is identity and that $G(z) \in [-1, 1]$, we reparameterize $G$ by $G(z) = 2 \cdot H(z) + z$ where the last activation layer of $G$ is tanh. we use similar architecture as the previous experiment for our model. Specifically, the architecture is FC(16*16 → 200)-BN-RELU-FC(200 → 500)-BN-RELU-FC(500 → 16*16)-Tanh for $H$ in the generator, and FC(16*16 → 200)-RELU-FC(200 → 500)-RELU-FC(500 → 1) for both $\phi$ and $\epsilon$ in the discriminator. We use the same optimizers and hyperparameters as experiment above.

### 8.10 ADDITIONAL FIGURES

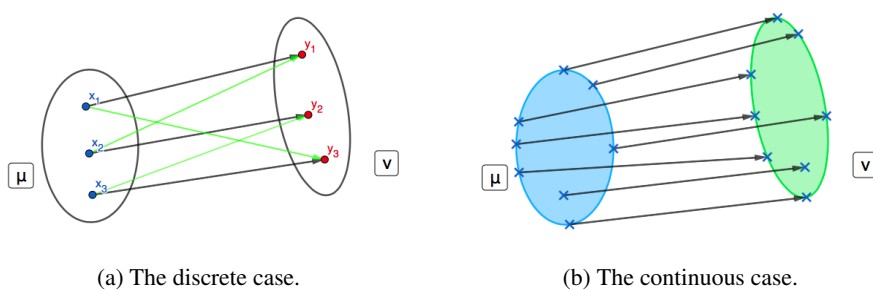

(a) The discrete case.      (b) The continuous case.

Figure 6: The Monge problem. **(a)** A discrete example of the Monge problem (1) for distributions in $\mathbb{R}^2$. The $\mu$ distribution consists in three equally weighted diracs in $x_1$, $x_2$ and $x_3$, while the $\nu$ one is represented by $y_1$, $y_2$ and $y_3$ in the same way. Black arrows denote the actual optimal transport map T. The green arrows together also define a map from $\mu$ onto $\nu$, but it is not optimal. **(b)** A continuous example of the Monge problem (1). $\mu$ and $\nu$ are uniform distributions on the blue and green ellipsoids respectively. The optimal transport map T is defined for any point in the support of $\mu$, and we see how it transports some points onto $\nu$'s support with the arrows.

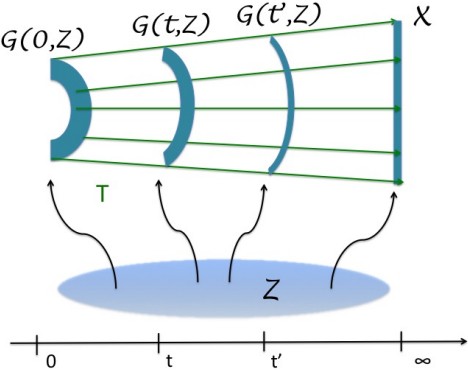

Figure 7: The time evolving generated distribution minimizing its Wasserstein distance with the true distribution $X$. The latent space is fixed and we denote it as $Z$. Green arrows give the shape of the optimal transport map $T$ between the initial distribution $G(0, Z)$ and the true distribution $X$. During training, $G(t, Z)$ does not follow an arbitrary path for converging toward $X$. It follows the Wasserstein-2 geodesic between $G(0, Z)$ and $X$ described by T.

