# OpenReview forum: "W2GAN: RECOVERING AN OPTIMAL TRANSPORT MAP WITH A GAN"
_ICLR.cc/2019/Conference_

### Official Review · AnonReviewer1 · 2018-10-31
**GANs for OT and OT for GANs**

**Rating:** 4
**Confidence:** 3

**Review:**

The paper proposes W2GAN, a GAN where the objective function relies on a W2 distance. Authors state that the discriminator approximate the W2 distance, and that the generator follows an OT map.
While I did not see any flaws in the development, the paper is quite bushy and hard to follow. Some questions are still open, for instance in the end of the experiments, authors state that the model has "a strong theoretical advantages": can you provide more details about those advantages?
The experiments do not show any clear advantages of the method regarding competitors. Regarding Table 1, why are there some points with no arrows? W2-OT seems not to perform better: are there some other advantages (computational?) to use the method? In Figure 1, it is quite difficult to evaluate the results on a single image with no comparisons. Again, providing a strong evaluation of the method would help to strengthen the paper.

There are some weird statements and typos mistakes that should be corrected. For example in the first 2 pages: (abstract) "other GANs also approximately following the Optimal Transport", (Introduction) "An optimal map has many important implications such as computing barycenters", "high-dimenisonal", "generator designed", "consideral", "although the theoretical arguments do not scale immediately".
The layout of the bibliography should be deeply reviewed.

---

> ### Author Response · Authors · 2018-11-23
> **Response to review**
>
> Thank you for your thorough and insightful review. Below we try to answer the questions you addressed.
>
> * authors state that the model has "a strong theoretical advantages": can you provide more details about those advantages?
>
> Theoretical advantages of W2GAN are the following:
> We can characterize the path the generator is following during training, namely the W2 geodesics. This is not the case for other GANs.
> A practical consequence of the above is that at the end of training, the generator is recovering an OT map.
>
>
> * The experiments do not show any clear advantages of the method regarding competitors
>
> In high dimensional data, we show that W2GAN outperforms the Barycentric-OT approach by (Seguy et al. 2018) in MV Gaussian to MNIST experiment. In 2D data, our method performs as well as other methods, but is simpler than the two-step approach of Barycentric-OT and has a stronger theoretical basis for recovering OT maps than WGAN and its extensions. As for the discrete method, which achieves perfect results in low dimension, do not scale to continuous and high-dimensional distributions.
>
>
> * Table 1: why are there some points with no arrows?
>
> We only visualize a fixed number (150) of mappings for the sake of clarity. We make this clearer in our updated version.
>
>
> * Table 1: W2-OT seems not to perform better: are there some other advantages (computational?) to use the method?
>
> Please check the general comment which clarifies this point.
>
>
> * In Figure 1, it is quite difficult to evaluate the results on a single image with no comparisons. Again, providing a strong evaluation of the method would help to strengthen the paper.
>
> We will provide a direct comparison with Barycentric-OT of (Seguy et al. 2018) for MNIST experiment in the updated version. As for CIFAR-10, please check our response to the same point raised by Reviewer-1.

---

> ### Author Response · Authors · 2018-12-03
> **Reconsidering Decision**
>
> Dear reviewer,
>
> We would appreciate that you reconsider your decision given the updated version of the paper.
>
> Thanks a lot.

---

### Official Review · AnonReviewer2 · 2018-10-31
**Original contribution unclear**

**Rating:** 3
**Confidence:** 4

**Review:**

The paper W2GAN describes a method for training GAN and computing an optimal transport (OT) map
between distributions. As far as I can tell, it is difficult to identify the original contributions
of the paper. Most results are known from the OT community. The differences with the work of Seguy, 2018
is also not obvious. I encourage the authors to establish more clearly the differences of their work
with this last reference. Most of the theoretical considerations of Section 3 is either based on
unrealistic assumptions (case 1) or make vague assumptions 'if we ignore the possibly significant effect ...'
that seem unjustified so far. Experimental results do not show evidences of superiority wrt. existing works.
All in all I would recommend the authors to better focus on the original contribution of their works wrt.
state-of-the-art and explain why the theoretical analysis on convergence following a geodesic path in a
Wasserstein space is valuable from a practical view. Finally, I did not understand the final claim of the
Abstract : 'Perhaps surprisingly, we also provide empirical evidence that other GANs also approximately following
the Optimal Transport.'. What are those empirical evidences ? It seems that this claim is not supported somewhere
else in the paper.

Minor remarks:
 - regarding the penalization in eq. (5), the expectation is not for all x and y \in R^2, but for x drawn from \mu and y from \nu.
   Same for L_2 regularization
 - Proposition 1 is mainly due to Brenier
Brenier, Y. (1991). Polar factorization and monotone rearrangement of vector‐valued functions. Communications on pure and applied mathematics, 44(4), 375-417.
 - from Eq (7), you should give precisely over what the expectations are taken.
 - Eq (10) : how do you inverse sup and inf ?
 - when comparing to Seguy 2018, are you using an entropic or a L_2 regularization ? How do you set the regularization strength ?
 - where is Figure 2.a described in section 4.2 ?

Related works :
 - what is reference (Alexandre, 2018) ?
 - regarding applications of OT to domain adaptation, there are several references on the subject.
   See for instance
Courty, N., Flamary, R., Tuia, D., & Rakotomamonjy, A. (2017). Optimal transport for domain adaptation. IEEE transactions on pattern analysis and machine intelligence, 39(9), 1853-1865.
or
Damodaran, B. B., Kellenberger, B., Flamary, R., Tuia, D., & Courty, N. (2018). DeepJDOT: Deep Joint distribution optimal transport for unsupervised domain adaptation. ECCV
for a deep variant.
 - Reference Seguy 2017 and 2018 are the same and should be fused. The corresponding paper
   was published at ICLR 2018
   Regarding this last reference, the claim 'As far as we know, it is the first demonstration of a GAN achieving reasonable generative modeling results and an approximation of the optimal transport map between two continuous distributions.' should maybe be lowered ?

---

> ### Author Response · Authors · 2018-11-23
> **Response to review - part 2**
>
> * did not understand the final claim of the empirical evidence that other GANs also approximately following the Optimal Transport
>
> In our 2D experiments, we find that WGAN_LP can find a mapping which is very close to the perfect OT generated by the discrete method. While we cannot make theoretical statements about the training dynamics in W1 GANs (because there are infinite W1 geodesics), it seems like in practice -- at least in our 2D experiments --  WGAN-LP seems to recover an OT map. Note that we observe that WGAN-GP does not work well in practice, which is in large part a result of forcing the discriminator’s gradient to be exactly 1, and hence distorting the local OT direction.
>
>
> * penalization in eq. (5), the expectation is not for all x and y \in R^2, but for x drawn from \mu and y from \nu.  Same for L_2 regularization and Eq (7).
>
> About eq. (5): this is true that the expectation should be taken over the marginals from the mere definition of the entropic (L2) regularized Kantorovitch problem. We corrected that.
> In the context of the unregularized Kantorovitch dual, the hard inequality constraint (eq (4)) should actually happen pointwise everywhere on the euclidean space. This is the objective we really want to approach in our context. In order to do that in a tractable manner, we remove the hard constraint and add a corresponding penalty in the objective of the discriminator. The perfect penalty should thus involve an expectation everywhere on the space. Thus one could try enforcing this penalty by sampling between the distributions, or around the distributions, etc. This applies in particular for the penalty term in eq (7).
>
>
> * Proposition 1 is mainly due to Brenier
>
> That is true, and it was not our intention to claim this result as ours. We will make this clearer in the updated version.
>
>
> * Eq (10) : how do you inverse sup and inf ?
>
> Thanks for noticing this. We will make this clearer in the next version.
>
>
> * when comparing to Seguy 2018, are you using an entropic or a L_2 regularization ? How do you set the regularization strength ?
>
> We use the L2 penalty. We find that in practice the entropic one in the dual results with an exponential signal, and eventually leads to divergence of the discriminator. We set the regularization strength with a hyper-parameter scalar.

---

> ### Author Response · Authors · 2018-11-23
> **Response to review - part 1**
>
> Thank you for your thorough and insightful review. Below we try to answer the questions you addressed.
>
> * It is difficult to identify the original contributions of the paper
>
> We took into account this consideration seriously. Please check the general comment in which we clarify our contributions. We will also make sure that the updated version of our paper states them clearly.
>
>
> * Most results are known from the OT community
>
> We agree that all the theoretical results from OT theory employed in this paper are well known for OT community. We borrow these results to propose a new model and analyse its behaviour during training.
>
>
> * The differences with the work of Seguy, 2018 is also not obvious
>
> The work of (Seguy et al., 2018) proposes a two-step approach for large-scale OT map. First, they solve the regularized Kantorovitch problem in its dual form to get an optimal *plan*, and then learn a parametric function (a neural network) to approximate the barycentric projection of this optimal plan. This is quite different from our approach where we use an adversarial approach where the discriminator locally approximates the optimal *map* toward the target distribution to provide signal for the generator. What should be compared when approaching the OT map is our generator at the end of training and their model after barycentric projection.
>
>
> * Most of the theoretical considerations of Section 3 is either based on unrealistic assumptions (case 1) or make vague assumptions 'if we ignore the possibly significant effect ...' that seem unjustified so far
>
> We acknowledge that the current analysis is based on idealistic assumptions, but these assumptions are not exclusive to our work and, in fact, most GAN papers make similar assumptions. This is a result of the difficulty of making concrete statements about parametric functions. However, in the updated version, we try to bridge the gap between the practical case of parametric update of the generator, and the idealistic case of continuous optimization in the space of non-parametric generator distribution. For this, we use functional gradient analysis, which helps in interpreting the parameter update as an approximation of an ideal discrete update in the space of probability measures.
>
>
> * why the theoretical analysis on convergence following a geodesic path in a     Wasserstein space is valuable from a practical view
>
> From a practical view, the main advantage is that this allows us to predict that the generator will recover an OT map at the end of training. This is interesting because:
> 1) In domain transfer applications (e.g. unsupervised image or language translation) it would be really valuable to characterize and possibly control the mapping obtained by a generative model
> 2) Large-scale Monge maps have also many practical applications (e.g. domain adaptation). Hence, having a powerful generative model approximating a Monge map can be powerful tool in these applications.
> Another possible practical application of following W2 geodesics between two distribution is that it allows us to observe intermediate probability measures, i.e. computing barycenters.
> That being said, we believe that characterizing dynamics of generator distribution to be theoretically very interesting and could possibly lead to advances in understanding GANs in general.

---

> ### Author Response · Authors · 2018-12-03
> **Reconsidering Decision**
>
> Dear reviewer,
>
> We would appreciate that you reconsider your decision given the updated version of the paper.
>
> Thanks a lot.

---

### Official Review · AnonReviewer3 · 2018-11-02
**Interesting approach to OT GAN for Wasserstein distances with regularised Kantorovitch duals**

**Rating:** 6
**Confidence:** 3

**Review:**


pros

- formal approach to the problem and a clear understanding of what is missing (Section 6.6); I appreciated Section 3 at large in particular.

- I like Theorem 1 and Corollary 1. Is it is possible to reason about an imperfect generator class and undertrained discriminator, and get sufficient conditions for convergence (not necessarily exponential) ?


cons

- In Proposition 1, I suspect that p > 2 (see below), which makes the p=2 choice a limit case of the proposition.

- The paper should have cited the paper https://arxiv.org/pdf/1710.05488.pdf which goes along similar lines in its Section 3 and make proper comparisons.

 - experimental results do not do a great favour to the technique proposed: in Table 1, W2-OT is not better than Barycentric-OT (see spiral); in Table 2, W2GAN is not better than WGAN-LP; Figure 1-a is maybe the only Figure with a clearcut advantage. However, the CIFAR examples in Figure 1b look quite bad after zooming. Do the authors have more experiments and comparisons on images ?

Detail:

* In proposition 1, (6), use the Holder conjugate of p: ||\nabla||^{1(p-1)-1} =1/||\nabla||^{2-q}. Also better to understand as $q\leq 2$.

* looking at the proof of proposition 1, I do not know how you derive the inverse gradient, but I suspect you need in fact $p>2$, which also implies $q<2$ above.

* Sentence after (10) grammatically incorrect

* In the interpretation of the equation after (16), isn’t is possible to interpret the Jacobian terms as a geometric tweak for the update of G ?

* Lots of mistakes in references: Mistake in the first ref in references, many @JOURNAL/CONF titles do not appear.

---

> ### Author Response · Authors · 2018-11-23
> **Response to review**
>
> Thank you for your thorough and insightful review. Below we try to answer the questions you addressed.
>
> * Regarding Theorem 1 and Corollary 1: Is it is possible to reason about an imperfect generator class and undertrained discriminator, and get sufficient conditions for convergence (not necessarily exponential) ?
>
> This is a very interesting point. Indeed, assuming a first bound on the difference between the gradient of our discriminator and the one of the perfect Kantorovitch potential, and a second bound on the generator update, one can compose those bounds to obtain, locally at one update, a bound on the deviation from the OT trajectory. It might be interesting then to add those bouds together (with some stochasticity on the direction of the error term) to see at the end of training how far the generated distribution is from the Monge map. In addition, an interesting as well as a hard problem is to ensure that the gradient of the regularized Kantorovitch potential converges toward the gradient of the Kantorovitch potential when the regularization term goes to zero, and how fast it does so.
>
>
> * In Proposition 1, I suspect that p > 2 (see below), which makes the p=2 choice a limit case of the proposition
> * In proposition 1, (6), use the Holder conjugate of p: ||\nabla||^{1(p-1)-1} =1/||\nabla||^{2-q}. Also better to understand as $q\leq 2$.
> * looking at the proof of proposition 1, I do not know how you derive the inverse gradient, but I suspect you need in fact $p>2$, which also implies $q<2$ above
>
> Those three remarks are linked. We can indeed use the Holder conjugate, but we do not need p>2. When p=2, the term in the denominator is equal to 1 and we actually recover the famous version of this proposition : T(x)=x-\nabla \phi(x).
> Your question about the way of inverting the gradient is legitimate, we omitted the fact that we use the L2 norm to define the cost function. Using L2 allows both computing and inverting the gradient easily. The case of p=2 is actually the easiest one, because the gradient and its inverse are (a scalar times) the identity. Please note that this proposition is a well known result by (Brenier, 1991) and not our own contribution. We will make sure this is made more explicit in the next version of the paper.
>
>
> * In the interpretation of the equation after (16), isn’t is possible to interpret the Jacobian terms as a geometric tweak for the update of G ?
>
> In fact, it is really hard to understand the effect of the Jacobian term. We are not sure what you mean by geometric tweak, but this term should be thought of as a projection of the ideal update in the space of probability measures onto the space of parametrized probability measures. According to the functional gradient analysis (which will be available in the next version of the paper), this update is actually meant to be the best update in the parameter space so that the updated generated distribution is closest as possible to the ideal updated distribution.
>
>
> * Cite and compare with https://arxiv.org/pdf/1710.05488.pdf
>
> Thanks for pointing out this work. We will make sure to cite it in the updated version of our paper. That being said, we believe it is quite different from our work. Their approach devices a generative model which uses an encoder-decoder process to reduce data to a latent space, and then perform the OT with a discriminator-only method (similar to our discriminator) in this latent space. This relies on the fact that T(x)=x-\nabla\phi might provide a good approximation of OT map in low dimensional spaces. Crucially, their generative model does not recover an optimal map between the two distributions in the data space, which is what our approach aims to achieve.
>
>
> * W2-OT is not better than Barycentric-OT (e.g. spirals)
>
> As stated in the general comment to all reviewers, the approximated OT by the discriminator is not meant to be a competitive approach on its own. We use it to provide the correct direction for the generator to be updated during training. The generator, on the other hand, is competitive to other large-scale OT approaches (such as Barycentric-OT).
>
>
> * W2GAN is not better than WGAN-LP
>
> We agree that WGAN-LP seems to perform very well in practice. The main advantage of W2GAN is that it relies on the Wasserstein-2 distance, which allows for a concrete theoretical analysis of its behaviour. This is not possible with Wasserstein-1 distance -- as explained in the comment to all reviewers.
>
>
> * CIFAR-10 experiments
>
> The main objective of our approach is to show that it is a competitive method for large-scale OT. The main point of this experiment is to show that our method can also be a reasonable GAN-based generative model, but it is not meant to show that we can achieve state-of-the-art results with it. We agree that this was not clear from our submission, and we will try to emphasize that in our updated one.

---

> ### Author Response · Authors · 2018-12-03
> **Reconsidering decision**
>
> Dear reviewer,
>
> We would appreciate that you reconsider your decision given the updated version of the paper.
>
> Thanks a lot.

---

### Author Response · Authors · 2018-11-23
**General response to all reviewers**

First, we would like to thank all reviewers for their feedback and thoughtful comments. We acknowledge that the submitted version of the paper had many limitations in terms of clarity of the message and writing. We will fix this in the next version which will plan to submit by Monday.

However, we take this opportunity to emphasize the main contributions of our paper:

- We introduce W2GAN, a GAN-based model which can compute optimal transport map in large-scale settings.

- We provide a theoretical analysis of training dynamics of the generator of W2GAN, which results in showing that the generator recovers the optimal transport map between the initial generator distribution and the real target distribution. Our analysis relies on two facts: the discriminator locally approximates the optimal transport map, and that it provides signal for the generator which allows it to follow the W2 geodesics during training.

- We provide empirical evidence that the generator recovers the optimal transport map between two distributions in both low-dimensional settings and high dimensional settings (MV Gaussians to MNIST).

We also want to address general concerns raised by the reviewers:

- The discriminator (or W2-OT) is not meant to be a competitive way to compute a Monge map. That was not clear in the first version of the paper. The discriminator is a real function and is not explicitly trained to reproduce a Monge map. What is important is that its gradient orients the generator toward the optimal map. Experiments in Table 1 should be seen as a confirmation of this theoretical fact.

- We do not aim to introduce a state-of-the-art GAN model. We regard our model as a principled method for computing OT map, which can scale for high-dimensional data. This is why we do not focus on achieving state-of-the-art results on CIFAR10. The main objective of this result was to show that our model, as opposed to other large-scale OT approaches, can achieve performance comparable to GAN in high-dimensional datasets.

- Why use W2 instead of W1?
Our theoretical argument that the generator recovers an optimal map at the end of training relies on analysing the path that it is following during the training. In the W2 case, this path is the unique Wasserstein 2 geodesic between two distributions. Hence, at the end of training, the generator recovers a Monge map. In the W1 case, it might be --although not in an obvious way-- that the generator follows a Wasserstein 1 geodesic, but those geodesics can be infinite. So checking that they correspond to a certain Monge map is not guaranteed, neither is the uniqueness of such map.

---

### Author Response · Authors · 2018-11-26
**Updated version of the paper**

Dear reviewers, we posted a new version of our paper, which includes the following modifications:

1- We clarified the message of our paper, our contributions, and improved the writing of the paper in general.

2- We provided a clearer analysis of the theoretical result we claim, i.e that the generator recovers an optimal map at the end of training.

3- We improved our experimental section, by clarifying our contributions in low dimensional data. We also included a new set of experiments in high dimensional data, where we applied our model to unsupervised domain adaptation and show that it can obtain competitive results. We removed CIFAR-10 experiment, as we realized it does not highlight the main contribution of this paper. Finally, we added low dimensional experiments confirming the theoretical analysis about the evolution of the generated distribution during training.

---

### Meta-Review · Area_Chair1 · 2018-12-14
**no clear practice advantage**

**Confidence:** 4
**Recommendation:** Reject

**Metareview:**

The paper introduces a W2GAN method for training GAN by minimizing 2-Wasserstein distance using
by computing an optimal transport (OT) map between distributions. However, the difference of previous works  is not significant or clearly clarified as pointed out some of the reviewers. The advantage of W2GAN over standard WGAN is also superficially explained, and did not supported by strong empirical evidence.